# Deep Molecular Representation Learning via Fusing Physical and Chemical Information

**Shuwen Yang[1]\*, Ziyao Li[2]\*, Guojie Song[1]†, Lingsheng Cai[1]**
[1]Key Laboratory of Machine Perception and Intelligence (MOE),
Peking University, Beijing, China
[2]Center for Data Science, Peking University, Beijing, China
{swyang,leeeezy,gjsong,cailingsheng}@pku.edu.cn

## Abstract

Molecular representation learning is the first yet vital step in combining deep learning and molecular science. To push the boundaries of molecular representation learning, we present PhysChem, a novel neural architecture that learns molecular representations via fusing physical and chemical information of molecules. PhysChem is composed of a physicist network (PhysNet) and a chemist network (ChemNet). PhysNet is a neural physical engine that learns molecular conformations through simulating molecular dynamics with parameterized forces; ChemNet implements geometry-aware deep message-passing to learn chemical / biomedical properties of molecules. Two networks specialize in their own tasks and cooperate by providing expertise to each other. By fusing physical and chemical information, PhysChem achieved state-of-the-art performances on MoleculeNet, a standard molecular machine learning benchmark. The effectiveness of PhysChem was further corroborated on cutting-edge datasets of SARS-CoV-2.

## 1 Introduction

The intersection between deep learning and molecular science has recently caught the eye of researchers in both areas. Remarkable progress was made in applications including molecular property prediction [36, 16], molecular graph generation [37, 12, 28], and virtual screening for drug discovery [8, 31], yet learning representations for molecules remains the first yet vital step. Molecular representation learning, or learning molecular fingerprints, aims to encode input notations of molecules into numerical vectors, which later serve as features for downstream tasks. Earlier deep molecular representation methods generally used *off-the-shelf* network architectures including message-passing neural networks [9], graph attention networks [36] and Transformers [11, 24]. These methods took either line notations (e.g. SMILES[3]) or graph notations (i.e. structural formulas) of molecules as inputs, whereas physical and chemical essence of molecules was largely neglected. Notably, a trend of integrating 3D *conformations* (i.e. the 3D Cartesian coordinates of atoms) into molecular representations recently emerged [4, 15, 29], while most of these methods assume the availability of labeled conformations of target molecules.

In order to push the boundaries of molecular representation learning, we revisited molecules from both *physical* and *chemical* perspectives. Modern physicists generally regard molecules as particle systems that continuously move following the laws of (quantum) mechanics. The dominant conformations of molecules reflect the *equilibriums* of these micro mechanical systems, and are thus of wide interest.

---

\*Equal Contribution.

†Corresponding Author.

[3]SMILES (Simplified Molecular Input Line Entry Specification [34]) is a widely used protocol that specifies (non-unique) line notations for molecules, `CCO` for ethanol, for example.

Chemists, on the other hand, focus more on *chemical bonds* and *functional groups* of molecules, which denote the interactions of electrons and determine chemical / biomedical properties such as solubility and toxicity, etc. Nevertheless, physical and chemical information of molecules is not orthogonal. For example, torsions of bond lengths and angles greatly influence the dynamics of particle systems. Therefore, an ideal molecular representation is not only expected to capture both physical and chemical information, but also to appropriately fuse the two types of information.

Based on the observations above, we propose **PhysChem**, a novel neural architecture that captures and fuses physical and chemical information of molecules. PhysChem is composed of two *specialist networks*, namely a *physicist network* (PhysNet) and a *chemist network* (ChemNet), who *understand* molecules physically and chemically.[4] **PhysNet** is a neural physical engine that learns dominant conformations of molecules via simulating molecular dynamics in a generalized space. In PhysNet, implicit positions and momenta of atoms are initialized by encoding input features. Forces between pairs of atoms are learned with neural networks, according to which the system moves following laws of classic mechanics. Final positions of atoms are supervised with labeled conformations under spatial-invariant losses. **ChemNet** utilizes a message-passing framework [9] to capture chemical characteristics of atoms and bonds. ChemNet generates messages from atom states and local geometries, and then updates the states of both atoms and bonds. Output molecular representations are merged from atomic states and supervised with labeled chemical / biomedical properties. Besides focusing on their own *specialties*, two networks also *cooperate* by sharing *expertise*: PhysNet consults the hidden representations of chemical bonds in ChemNet to generate torsion forces, whereas ChemNet leverages the local geometries of the intermediate conformations in PhysNet.

Compared with existing methods, PhysChem adopts a more elaborated as well as interpretable architecture that incarnates physical and chemical understandings of molecules. Moreover, as PhysNet learns molecular conformations from scratch, PhysChem does not require labeled conformations of test molecules. This extends the applicability of PhysChem to situations where such labels are unavailable, for example, with neural-network-generated drug candidates. We evaluated PhysChem on several datasets in the MoleculeNet [35] benchmark, where PhysChem displayed state-of-the-art performances on both conformation learning and property prediction tasks. Results on cutting-edge datasets of SARS-CoV-2 further proved the effectiveness of PhysChem.

## 2 Related Work

**Molecular Representation Learning** Early molecular fingerprints commonly encoded line or graph notations of molecules with rule-based algorithms [19, 22, 25]. Along with the spurt of deep learning, deep molecular representations gradually prevailed [7, 9, 11, 36]. More recently, researchers started to focus on incorporating 3D conformations of molecules into their representations [1, 4, 29, 15]. Models that leveraged 3D geometries of molecules generally performed better than those that simply used graph notations, whereas most 3D models required labeled conformations of the target molecules. This limited the applicability of these models. Among previous studies, message-passing neural networks (MPNNs) proposed a universal framework of encoding molecular graphs, which assumed that nodes in graphs (atoms in molecules) passed messages to their neighbors, and then aggregated received messages to update their states. A general message-passing layer calculated

$$\boldsymbol{m}_i = \sum_{j \in \mathcal{N}(v)} M(\boldsymbol{h}_i, \boldsymbol{h}_j, \boldsymbol{e}_{i,j}), \quad \boldsymbol{h}_i \leftarrow U(\boldsymbol{h}_i, \boldsymbol{m}_i), \quad i \in \mathcal{V}, \tag{1}$$

where $i, j \in \mathcal{V}$ were graph nodes, $\boldsymbol{h}$s and $\boldsymbol{e}$s were states of nodes and edges, and $M(\cdot), U(\cdot)$ were the *message* and *update* functions. For graph-level tasks, MPNNs further defined a *readout function* which merged the states of all nodes into graph representations. In this work, ChemNet modifies the message-passing scheme of MPNNs: messages are extended to incorporate geometries of local structures, and states of both atoms and bonds are updated. See Section 3.4 for more details.

**Neural Physical Engines** Recent studies showed that neural networks are capable of learning annotated (or pseudo) potentials and forces in particle systems, which made fast molecular simulations [38, 17] and protein-folding tasks [26] possible. Notably, it was further shown in [16, 27] that neural networks alone can simulate molecular dynamics for conformation prediction. As an instance,

---

[4]PhysNet and ChemNet are novel architectures, not to be confused with previous works with similar or identical names [33, 32, 20].

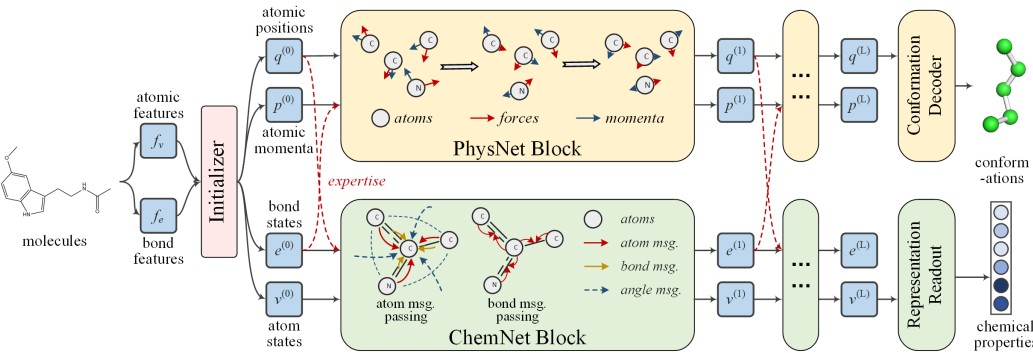

Figure 1: A sketch of the architecture of PhysChem.

HamNet [16] proposed a neural physical engine that operated on a generalized space, where positions and momentums of atoms were defined as high-dimensional vectors. In the engine, atoms moved following Hamiltonian Equations with parameterized kinetic, potential and dissipation functions. PhysNet in our work is a similar engine. Nevertheless, instead of learning parameterized energies and calculating their negative derivatives as forces, we directly parameterize the forces between each pair of atoms. In addition, HamNet considered gravitations and repulsions of molecules based on implicit positions, while it ignored the effects of chemical interactions: for example, the types of chemical bonds were ignored in the energy functions. PhysChem fixes this issue via the cooperation mechanism between two specialist networks. Specifically, PhysNet takes chemical expertise (the bond states) from ChemNet and introduces *torsion forces*, i.e. forces that origin from torsions in chemical bonds, into the dynamics. See Section 3.3 for more details.

**Multi-Task Learning and Model Fusion** The cooperation mechanism in PhysChem shares similar motivations with *multi-task learning* and *model fusion*. Multi-task learning [39, 6] is now almost universally used in deep learning models. Representations are shared among a collection of related tasks in order to learn the common ideas. Model fusion, on the other hand, merges different models on identical tasks to improve performances [23, 13]. Notably, these techniques have been previously applied to molecular tasks [30, 13, 31]. In PhysChem, conformation learning and property prediction tasks are jointly trained, with two specialist networks fused to achieve better performances. Nevertheless, the cooperation mechanism in PhysChem roots from observations in physics and chemistry, and enjoys better interpretability than straight-forward ensemble strategies. The advantages of the cooperation mechanism over multi-task strategies are also empirically shown in Section 4.

## 3 Proposed Method: PhysChem

### 3.1 Preliminaries

**Notations** In the statements and equations below, we use italic letters for scalars and indices, bold lower-case letters for (column) vectors, bold upper-case letters for matrices, calligraphic letters for sets, and normal letters for annotations. Common neural network layers are directly referred to: FC $(\boldsymbol{x})$ denotes a fully-connected layer with input $\boldsymbol{x}$; GCN $(\boldsymbol{A}, \boldsymbol{X})$ denotes a (vanilla) graph convolutional network (GCN)[14] with adjacency matrix $\boldsymbol{A}$ and feature matrix $\boldsymbol{X}$; $\mathrm{GRU}_{\mathrm{cell}}(\boldsymbol{s}, \boldsymbol{x})$ denotes a cell of the Gated Recurrent Units (GRU) [5] with recurrent state $\boldsymbol{s}$ and input signal $\boldsymbol{x}$; LSTM $(\{\boldsymbol{x}_t\})$ denotes a Long Short-Term Memory network [10] with input signals $\{\boldsymbol{x}_t\}$; MaxPool $(\{\boldsymbol{x}_t\})$ denotes a max-pooling layer with inputs $\{\boldsymbol{x}_t\}$. Exact formulas are available in the appendix. $\oplus$ is the operator of concatenations. $\|\cdot\|$ denotes the $l_2$ norm of the input vector.

**Problem Definition** PhysChem considers molecular representation learning as a supervised learning task. It takes notations of molecules as inputs, conformations and chemical / biomedical properties as supervisions. PhysChem assumes a pre-conducted featurization process, after which a molecule can be denoted as an attributed graph $\mathcal{M} = (\mathcal{V}, \mathcal{E}, n, m, \boldsymbol{X}^{\mathrm{v}}, \boldsymbol{X}^{\mathrm{e}})$. Here, $\mathcal{V}$ is the set of $n$ atoms, $\mathcal{E} \subset \mathcal{V} \times \mathcal{V}$ is the set of $m$ chemical bonds, $\boldsymbol{X}^{\mathrm{v}} \in \mathbb{R}^{n \times d_{\mathrm{v}}} = (\boldsymbol{x}_1^{\mathrm{v}}, \cdots, \boldsymbol{x}_n^{\mathrm{v}})^{\top}$ is the matrix of atomic

features, and $\boldsymbol{X}^{\mathrm{e}} \in \mathbb{R}^{m \times d_{\mathrm{e}}} = (\boldsymbol{x}_1^{\mathrm{e}}, \cdots, \boldsymbol{x}_m^{\mathrm{e}})^\top$ that of bond features. Based on above inputs, PhysNet outputs the dominant conformations of molecules, and ChemNet outputs the representations as well as the predicted chemical / biomedical properties of molecules.

**Overview** Figure 1 is a sketch of PhysChem. An *initializer* first encodes the inputs into initial atom and bond states ($\boldsymbol{v}^{(0)}$ and $\boldsymbol{e}^{(0)}$) for ChemNet, along with the initial atomic positions and momenta ($\boldsymbol{q}^{(0)}$ and $\boldsymbol{p}^{(0)}$) in PhysNet. Subsequently, $L$ PhysNet blocks simulate neural molecular dynamics in the generalized space; $L$ ChemNet blocks conduct geometry-aware message-passing for atoms and bonds. Between each couple of blocks, implicit conformations ($\boldsymbol{q}$s) and states of chemical bonds ($\boldsymbol{e}$s) are shared as expertise. At the top of PhysNet, a *conformation decoder* transforms implicit atomic positions into the 3D Euclidean space; of ChemNet, a sequence of *readout layers* aggregate atom states into molecular representations, based on which molecular properties are predicted with task-specific layers. Notably, two specialist networks are jointly optimized in PhysChem.

### 3.2 Initializer

Given a molecular graph $\mathcal{M} = (\mathcal{V}, \mathcal{E}, n, m, \boldsymbol{X}^{\mathrm{v}}, \boldsymbol{X}^{\mathrm{e}})$, the initializer generates the initial values for both PhysNet and ChemNet variables. In the initializer, we first encode the input features into initial atom and bond states with fully connected layers, i.e.

$$\boldsymbol{v}_i^{(0)} = \mathrm{FC}\left(\boldsymbol{x}_i^{\mathrm{v}}\right), \; i \in \mathcal{V}; \qquad \boldsymbol{e}_{i,j}^{(0)} = \mathrm{FC}\left(\boldsymbol{x}_{i,j}^{\mathrm{e}}\right), \; (i,j) \in \mathcal{E}. \tag{2}$$

We then adopt the initialization method in [16] to generate initial positions ($\boldsymbol{q}^{(0)}$) and momenta ($\boldsymbol{p}^{(0)}$) for atoms: a bond-strength adjacency matrix $\boldsymbol{A} \in \mathbb{R}^{n \times n}$ is estimated with sigmoid-activated FC layers on bond features, according to which a GCN captures the chemical environments of atoms (as $\tilde{\boldsymbol{v}}$); an LSTM then determines unique positions for atoms, especially for those with identical chemical environments (carbons in benzene, for example). Denoted in formula, the initialization follows

$$\boldsymbol{A}(i,j) = \begin{cases} 0, & (i,j) \notin \mathcal{E} \\ \mathrm{FC}_{\mathrm{sigmoid}}\left(\boldsymbol{x}_{i,j}^{\mathrm{e}}\right), & (i,j) \in \mathcal{E} \end{cases}, \qquad \tilde{\boldsymbol{V}} = \mathrm{GCN}\left(\boldsymbol{A}, \boldsymbol{V}^{(0)}\right), \tag{3}$$

$$\left\{\left(\boldsymbol{q}_i^{(0)} \oplus \boldsymbol{p}_i^{(0)}\right)\right\} = \mathrm{LSTM}\left(\{\tilde{\boldsymbol{v}}_i\}\right), \quad i \in \mathcal{V} \tag{4}$$

where $\boldsymbol{V}^{(0)} = \left(\boldsymbol{v}_1^{(0)}, \cdots, \boldsymbol{v}_n^{(0)}\right)^\top$ and $\tilde{\boldsymbol{V}} = (\tilde{\boldsymbol{v}}_1, \cdots, \tilde{\boldsymbol{v}}_n)^\top$ are the atom states. The order of atoms in the LSTM is specified by the canonical SMILES of the molecule.

### 3.3 PhysNet

Overall, PhysNet simulates the dynamics of atoms in a generalized, $d_f$-dimensional space ($d_f \geq 3$). In a molecule as a classic dynamical system, atoms move following Newton's Second Law:

$$\mathrm{d}\boldsymbol{q}/\mathrm{d}t = \boldsymbol{p}/m, \qquad \mathrm{d}\boldsymbol{p}/\mathrm{d}t = \boldsymbol{f}, \tag{5}$$

where $\boldsymbol{q}$, $\boldsymbol{p}$ and $m$ are the position, momentum and mass of an atom, and $\boldsymbol{f}$ is the force that the atom experiences. With uniform temporal discretization, the above equations may be approximated with

$$\boldsymbol{q}_{s+1} = \boldsymbol{q}_s + \boldsymbol{p}_s \tau/m, \qquad \boldsymbol{p}_{s+1} = \boldsymbol{p}_s + \boldsymbol{f}_s \tau, \qquad s = 0, 1, 2, \cdots \tag{6}$$

where $s$ is the index of timestamps and $\tau$ is the temporal interval. The calculations in PhysNet simulate such a process. Parameters of PhysNet blocks learn to derive the forces $\boldsymbol{f}$ from intermediate molecular conformations and states of chemical bonds. Correspondingly, two types of forces are modeled in PhysNet, namely the *positional forces* and the *torsion forces*.

**Positional Forces** The positional forces in PhysNet model the gravitations and repulsions between pairs of atoms. In conventional molecular force fields, these forces are generally modeled with (negative) derivatives of *Lennard-Jones potentials*. We therefore propose a similar form for the positional forces taking atomic distances as determinants:

$$\boldsymbol{f}_{j,i}^{\mathrm{pos}} = (r_{j,i}^{-2} - r_{j,i}^{-1}) m_j m_i \boldsymbol{d}_{j,i}, \quad r_{j,i} = \|\mathrm{FC}\left(\boldsymbol{q}_i - \boldsymbol{q}_j\right)\|, \quad i, j \in \mathcal{V}, \quad i \neq j, \tag{7}$$

where $\boldsymbol{d}_{j,i} = \frac{\boldsymbol{q}_i - \boldsymbol{q}_j}{\|\boldsymbol{q}_i - \boldsymbol{q}_j\|}$ is the unitary directional vector from atom $j$ to $i$. Instead of using $l_2$ distances in the generalized space, we use parameterized *interaction distances* $r_{j,i}$ to estimate the forces, which

increase the capability of the network. Here, $r^{-2} - r^{-1}$ approximates the landscape of derivatives of Lennard-Jones potentials[5] with lower-degree polynomials. The approximation is made to avoid numerical issues of using high-degree polynomials, most typically, the gradient explosion.

**Torsion Forces** The torsion forces model the mechanical effects of torsions in local structures. The torsion forces are defined between pairs of atoms that are directly connected by chemical bonds. Estimating local torsions only with positions and momenta of atoms is somehow suboptimal, as the network lacks prior knowledge of chemical characteristics of the bonds (default bond lengths, for example). Alternatively, we leverage the bond states ($e_{i,j}$) in ChemNet as chemical expertise to derive the torsion forces. Specifically, the torsion forces $f^{\text{tor}}$ are calculated as

$$ \boldsymbol{f}_{j,i}^{\text{tor}} = w_{j,i}\boldsymbol{d}_{j,i}, \quad w_{j,i} = \text{FC}\left(\boldsymbol{e}_{j,i}\right), \quad (i,j) \in \mathcal{E}. \tag{8} $$

As the torsion forces model chemical interactions, we do not explicitly incorporate atomic mass into the calculation. Notably, atomic information is implicitly considered in the torsion forces, as the bond states integrate atom states at both ends in ChemNet (see the next subsection).

**Dynamics** After estimating the positional and torsion forces, PhysNet simulates the dynamics of atoms following Equation (6). In the $l$-th block of PhysNet, $S$ steps of dynamics are simulated:

$$ \boldsymbol{q}_i^{(l,s+1)} = \boldsymbol{q}_i^{(l,s)} + \frac{\tau}{m_i}\boldsymbol{p}_i^{(l,s)}, \quad \boldsymbol{p}_i^{(l,s+1)} = \boldsymbol{p}_i^{(l,s)} + \tau\boldsymbol{f}_i^{(l,s)}, \quad s = 0, 1, \cdots, S-1. \tag{9} $$

Here, $\boldsymbol{q}_i^{(l,0)} = \boldsymbol{q}_i^{(l)}, \boldsymbol{q}_i^{(l+1)} = \boldsymbol{q}_i^{(l,S)}$ ($\boldsymbol{p}$ similarly). $\boldsymbol{f}_i$ is the resultant force on atom $i$, i.e.

$$ \boldsymbol{f}_i = \sum_{j \in \mathcal{V}} \boldsymbol{f}_{j,i}^{\text{pos}} + \sum_{(i,j) \in \mathcal{E}} \boldsymbol{f}_{j,i}^{\text{tor}}. \tag{10} $$

**Conformation Decoder and Loss** We use a simple linear transformation to decode implicit atomic positions (and momenta) from the generalized space into the real 3D space:

$$ \hat{\boldsymbol{R}} = \boldsymbol{Q}\boldsymbol{W}_{\text{dec}}, \quad \boldsymbol{W}_{\text{dec}} \in \mathbb{R}^{d_f \times 3}, \tag{11} $$

where $\boldsymbol{Q} = (\boldsymbol{q}_1, \cdots, \boldsymbol{q}_n)^\top \in \mathbb{R}^{n \times d_f}$ is the position matrix in the generalized space, and $\hat{\boldsymbol{R}} = (\hat{\boldsymbol{r}}_1, \cdots, \hat{\boldsymbol{r}}_n)^\top \in \mathbb{R}^{n \times 3}$ that of the predicted 3D conformation. We further propose Conn-$k$ ($k$-hop connectivity loss), a spatially invariant loss that supervises PhysNet based on local distance errors: if $\boldsymbol{C}^{(k)} \in \{0,1\}^{n \times n}$ is the $k$-hop connectivity matrix[6] and $\boldsymbol{D}$ is the distance matrix[7], then the $k$-hop connectivity loss is defined as

$$ L_{\text{Conn-}k}(\hat{\boldsymbol{R}}, \boldsymbol{R}) = \left\| \frac{1}{n}\hat{\boldsymbol{C}}^{(k)} \odot \left(\hat{\boldsymbol{D}} - \boldsymbol{D}\right) \odot \left(\hat{\boldsymbol{D}} - \boldsymbol{D}\right) \right\|_F, \tag{12} $$

where $\odot$ is the element-wise product, $\|\cdot\|_F$ is the Frobenius norm, $(\boldsymbol{D}, \hat{\boldsymbol{D}})$ are distance matrices of the real and predicted conformations $(\boldsymbol{R}, \hat{\boldsymbol{R}})$, and $\hat{\boldsymbol{C}}^{(k)}$ is the normalized $k$-hop connectivity matrix[8]. The total loss of PhysNet is defined as the weighted sum of Conn-3 losses on all timestamps:

$$ L_{\text{phys}} = \frac{1}{Z}\sum_{l,s} \eta^{LS-(lS+s)}L_{\text{Conn-3}}^{(l,s)}, \quad l = 1, 2, \cdots, L, \quad s = 1, 2, \cdots, S, \tag{13} $$

where $\eta < 1$ is a decay factor and $Z = 0.01$ is an empirical normalization factor.

### 3.4 ChemNet

ChemNet conducts message-passing for both atoms and chemical bonds. Besides generating messages with atom states, ChemNet also considers local geometries including bond lengths and angles to adequately characterize local chemical environments. Specifically, in each ChemNet block, *triplet descriptors* are established in the atomic neighborhoods. Atoms merge relevant triplet descriptors to

---

[5] $U^{\text{L-J}} = ar^{-12} - br^{-6}$, $\quad \boldsymbol{f}^{\text{L-J}} = -\frac{\partial U^{\text{L-J}}}{\partial \boldsymbol{x}} = \left(12ar^{-11} - 6br^{-5}\right)\frac{\partial r}{\partial \boldsymbol{x}}$

[6] $\boldsymbol{C}_{i,j}^{(k)} = 1$ if and only if atoms $i$ and $j$ are $k$- or less-hop connected on the molecular graph.

[7] $\boldsymbol{D}_{i,j} = \|\boldsymbol{r}_i - \boldsymbol{r}_j\|$, where $\boldsymbol{r} \in \mathbb{R}^3$ are 3D coordinates of atoms.

[8] $\hat{\boldsymbol{C}}^{(k)} = \boldsymbol{L}^{-1/2}\boldsymbol{C}^{(k)}\boldsymbol{L}^{-1/2}$, where $\boldsymbol{L}$ is the diagonal degree matrix with $\boldsymbol{L}_{i,i} = \sum_j \boldsymbol{C}_{i,j}^{(k)}$.

generate messages, and then aggregate received messages to update their states. Chemical bonds also update their states by aggregating the states of atoms at both ends.

**Triplet Descriptor** The triplet descriptors $\boldsymbol{t}_{i,j,k}$ are descriptive vectors defined on all atom triplets $(i, j, k)$ with $(i, j) \in \mathcal{E} \wedge (i, k) \in \mathcal{E}$:

$$\boldsymbol{t}_{i,j,k}^{(l)} = \text{FC}\left(\boldsymbol{v}_i^{(l)} \oplus \boldsymbol{v}_j^{(l)} \oplus \boldsymbol{v}_k^{(l)} \oplus \boldsymbol{l}_{i,j}^{(l)} \oplus \boldsymbol{l}_{i,k}^{(l)} \oplus \boldsymbol{a}_{j,i,k}^{(l)}\right), \quad j, k \in \mathcal{N}(i), \tag{14}$$

where $\mathcal{N}(i) = \{j \mid (i, j) \in \mathcal{E}\}$ denotes the atomic neighborhood of $i$, $\boldsymbol{v}$s are atom states, and $\boldsymbol{l}_{i,j} = \text{FC}\left(l_{i,j}\right)$ and $\boldsymbol{a}_{j,i,k} = \text{FC}\left(\cos\left(\angle_{j,i,k}\right)\right)$ are representations of bond lengths and bond angles. The motivation of constructing triplet descriptors is that these features are *geometrically deterministic and invariant*: i) all bond lengths and angles in an atomic neighborhood together compose a minimum set of variables that uniquely determine the geometry of the neighborhood (*deterministic*); ii) these features all enjoy translational and rotational invariances (*invariant*). When PhysNet and ChemNet are jointly optimized, $l_{i,j}^{(l)}$ and $\angle_{j,i,k}^{(l)}$ are calculated from the intermediate 3D conformation in the $l$-th PhysNet block; when real conformations of target molecules are available, these values can be replaced by the ground-truths (e.g. ChemNet (*real conf.*) in Section 4).

**Message-passing** After establishing the triplet descriptors, ChemNet generates atomic messages by merging relevant descriptors. The message from atom $j$ to $i$ in the $l$-th block is calculated as

$$\boldsymbol{m}_{i,j}^{(l)} = \text{MaxPool}\left(\left\{\boldsymbol{t}_{i,j,k}^{(l)} \mid k \in \mathcal{N}(i)\right\}\right), \quad j \in \mathcal{N}(i), \quad l = 1, 2, \cdots, L. \tag{15}$$

Subsequently, ChemNet utilizes a similar architecture to [36] to conduct message passing. Centric atoms aggregate the received messages with attention scores determined by the bond states:

$$\boldsymbol{m}_i^{(l)} = \sum_{j \in \mathcal{N}(i)} \alpha_{i,j}^{(l)} \boldsymbol{m}_{i,j}^{(l)}, \quad \left\{\alpha_{i,j}^{(l)} \mid j \in \mathcal{N}(i)\right\} = \text{softmax}\left(\left\{\text{FC}\left(\boldsymbol{e}_{i,j}^{(l)}\right) \mid j \in \mathcal{N}(i)\right\}\right). \tag{16}$$

Atom states are then updated with GRU cells that take previous states as recurrent states. A similar process is then conducted for all chemical bonds, where messages are generated by the states of atoms at both ends. Denoted in formula,

$$\boldsymbol{v}_i^{(l+1)} = \text{GRU}_{\text{cell}}\left(\boldsymbol{v}_i^{(l)}, \boldsymbol{m}_i^{(l)}\right), \quad \boldsymbol{e}_{i,j}^{(l+1)} = \text{GRU}_{\text{cell}}\left(\boldsymbol{e}_{i,j}^{(l)}, \text{FC}\left(\boldsymbol{v}_i^{(l+1)} \oplus \boldsymbol{v}_j^{(l+1)}\right)\right). \tag{17}$$

**Representation Readout** The molecular representation is finally read out with $T$ global attentive layers [36], where a virtual *meta-atom* continuously collects messages from all atoms in the molecule and updates its state. We initialize and update the state of the meta-atom ($\boldsymbol{v}_{\text{meta}}$) following:

$$\boldsymbol{m}_{\text{meta}}^{(t)} = \sum_{i \in \mathcal{V}} \alpha_i^{(t)} \boldsymbol{v}_i^{(L)}, \quad \left\{\alpha_i^{(t)} \mid i \in \mathcal{V}\right\} = \text{softmax}\left(\left\{\text{FC}\left(\boldsymbol{v}_{\text{meta}}^{(t)} \oplus \boldsymbol{v}_i^{(L)}\right) \mid i \in \mathcal{V}\right\}\right),$$

$$\boldsymbol{v}_{\text{meta}}^{(0)} = \text{FC}\left(\frac{1}{n} \sum_{i \in \mathcal{V}} \boldsymbol{v}_i^{(L)}\right), \quad \boldsymbol{v}_{\text{meta}}^{(t+1)} = \text{GRU}_{\text{cell}}\left(\boldsymbol{v}_{\text{meta}}^{(t)}, \boldsymbol{m}_{\text{meta}}^{(t)}\right), \quad t = 1, 2, \cdots, T. \tag{18}$$

The final state of the meta-atom is used as the molecular representation. Labeled chemical and / or biomedical properties are used to supervise the representations, and the loss of ChemNet, $L_{\text{chem}}$, is determined task-specifically. The total loss of PhysChem is the weighted sum of losses in the two networks controlled by a hyperparameter $\lambda$, i.e.

$$L_{\text{total}} = \lambda L_{\text{phys}} + L_{\text{chem}}. \tag{19}$$

## 4 Experiments

### 4.1 Experimental Setup

**Datasets and Featurization** We evaluated PhysChem on quantum mechanics (QM7, QM8 and QM9) and physical chemistry (LIPOP, FREESOLV and ESOL) datasets, as well as drug effectiveness datasets of the notorious SARS-CoV-2. Statistics of the datasets are in Table 1. QM7, QM8 and QM9 contain stable organic molecules with up to 7, 8 or 9 *heavy* atoms. 3D atomic coordinates as

Table 1: Statistics of the datasets used in this paper.

| Datasets | QM7 | QM8 | QM9 | LIPOP | FREESOLV | ESOL | COVID19 |
|---|---|---|---|---|---|---|---|
| # molecules | 7,160 | 21,786 | 133,885 | 4,200 | 642 | 1,128 | 14,332 |
| # targets | 1 | 16 | 12 | 1 | 1 | 1 | 13 |
| Task type | Regression | | | Regression | | | Classification |
| Category | Quantum Mechanics | | | Physical Chemistry | | | Biomedicine |
| With conformation? | Yes | | | No | | | No |

well as electrical properties of molecules were calculated with *ab initio* Density Functional Theory (DFT). LIPOP provides experimental results on *lipophilicity* of 4,200 organic compounds; FREESOLV provides experimental and calculated hydration free energy of 642 small molecules in water; ESOL provides water solubility data for 1128 compounds. The above six datasets were collected in the MoleculeNet benchmark [35][9]. COVID19 [3] is a collection of datasets[10] generated by screening a panel of SARS-CoV-2-related assays against approved drugs. 13 assays of 14,332 drugs were used in our experiments. We split all datasets into 8:1:1 as training, validation and test sets. For datasets with less than $100,000$ molecules, we trained models for 5 replicas with randomly split data and reported the means and standard deviations of performances; for QM9, we used splits with the same random seed across models. We used identical featurization process in [36] to derive feature matrices $(\boldsymbol{X}^{\mathrm{v}}, \boldsymbol{X}^{\mathrm{e}})$ for all models and on all datasets.

**Baselines** For conformation learning tasks, we compared PhysChem with i) a Distance Geometry [2] method tuned with the Universal Force Fields (UFF) [21], which was implemented in the RDKit package[11] and thus referred to as RDKit; ii) CVGAE and CVGAE+UFF [18], which learned to generate low-energy molecular conformations with deep generative graph neural networks (either with or without UFF tuning); iii) HamEng [16], which learned stable conformations via simulating Hamiltonian mechanics with neural physical engines. For property prediction tasks, we compared PhysChem with i) MoleculeNet, for which we reported the best performances achieved by methods collected in [35] (before 2017); ii) 3DGCN [4], which augmented conventional GCNs with input bond geometries; iii) DimeNet [15], which conducted directional message-passing by representing pairs of atoms; iv) Attentive FP [36], which used local and global attentive layers to derive molecular representations; and v) CMPNN [29], which used communicative kernels to conduct deep message-passing. We conducted experiments with official implementations of HamEng, CVGAE,Attentive FP and CMPNN; for other baselines, as identical evaluation schemes were adopted, we referred to the reported performances in corresponding citations and left unreported entries blank.

**PhysChem Variants** We also compared PhysChem with several variants, including i) PhysNet (*s.a.*), a *stand-alone* PhysNet that ignored all chemical expertise by setting $\boldsymbol{e}_{j,i} \equiv \boldsymbol{0}$; ii) ChemNet (*s.a.*), a *stand-alone* ChemNet (ChemNet (*s.a.*)) that ignored all physical expertise by equally setting all bond lengths and angles ($l_{i,j} \equiv l, a_{j,i,k} \equiv a$); iii) ChemNet (*real conf.*) and ChemNet (*rdkit conf.*), two ChemNet variants that leveraged $l_{i,j}$ and $a_{j,i,k}$ in real conformations (*real conf.*) or RDKit-generated conformations (*rdkit conf.*); and iv) ChemNet (*m.t.*), a multi-task ChemNet variant that used a straight-forward multi-task strategy for conformation learning and property prediction tasks: we applied the *conformation decoder and loss* on atom states $\boldsymbol{v}$ and optimized ChemNet with the weighted sum of losses of both tasks (i.e. $L_{\mathrm{total}}$ in Equation (19)).

**Implementation and Training Details** Unless otherwise specified, we used $L = 2$ pairs of blocks for PhysChem. In the initializer, we used a 2-layer GCN and a 2-layer LSTM. In each PhysNet block, we set $d_f = 3$, $S = 4$ and $\tau = 0.25$. In each ChemNet block, we set the dimensions of atom and bond states as 128 and 64, correspondingly. In the representation readout block, we used $T = 1$ global attentive layers with 256-dimensional meta-atom states. For property prediction tasks with multiple targets, the targets were first standardized and then fitted simultaneously. We use the mean-square-error (MSE) loss for all regression tasks and the cross-entropy loss for all classification tasks to train the models. Other implementation details, such as hyperparameters of baselines, are provided in the appendix.

---

[9]The datasets are publicly available at `http://moleculenet.ai/datasets-1`.

[10]The datasets are available at `https://opendata.ncats.nih.gov/covid19/` (CC BY 4.0 license) and are continuously extended. The data used in our experiments were downloaded on February 16th, 2021.

[11]We use the 2020.03.1.0 version of the RDKit package at `http://www.rdkit.org/`.

Table 2: Performances of conformation learning on QM datasets. Results in more metrics are in the Appendix.

| Dataset | Metric | QM7 Distance MSE | QM8 Distance MSE | QM9 Distance MSE |
|---|---|---|---|---|
| single-task | Random Guess | 2.597±0.006 | 2.631±0.003 | 2.799 |
| | RDKit [2] | 1.236±0.011 | 1.635±0.006 | 1.920 |
| | CVGAE [18] | 1.110±0.011 | 1.053±0.007 | 1.052 |
| | CVGAE+UFF [18] | 1.109±0.003 | 1.049±0.008 | 1.048 |
| | HamEng [16] | 0.636±0.036 | 0.596±0.012 | 0.418 |
| | PhysNet (*s.a.*) | **0.504±0.013** | **0.257±0.006** | **0.197** |
| multi-task | ChemNet (*m.t.*) | 1.034±0.021 | 0.690±0.007 | 0.626 |
| | **PhysChem** | **0.492±0.027** | **0.259±0.008** | **0.255** |

Table 3: Performances of property prediction on QM datasets. W/ and w/o conf. specify whether conformations of test molecules were leveraged. *Italic* entries are directly referred to from citations. Individual MAEs for separate tasks on QM9 are listed in the Appendix.

| Dataset | Metric | QM7 MAE | QM8 Multi-MAE | QM9 Multi-MAE |
|---|---|---|---|---|
| w/o conf. | Random Guess | 178.8±6.4 | 0.0343±0.0025 | 9.392 |
| | MoleculeNet [35] | *94.7±2.7* | *0.0150±0.0020* | *2.350* |
| | Attentive FP [36] | 66.2±2.8 | 0.0130±0.0006 | *1.292* |
| | ChemNet (*s.a.*) | **59.6±2.0** | 0.0105±0.0002 | 1.209 |
| | ChemNet (*m.t.*) | 60.3±2.4 | 0.0112±0.0005 | 1.639 |
| | **PhysChem** | **59.6±2.3** | **0.0101±0.0003** | **1.096** |
| w/ conf. | DimeNet [15] | – | – | *1.920* |
| | HamNet [16] | – | – | *1.194* |
| | ChemNet (*rdkit conf.*) | **60.1±2.0** | 0.0100±0.0003 | 1.140 |
| | ChemNet (*real conf.*) | 60.2±1.9 | **0.0098±0.0003** | **1.040** |

## 4.2 Results

**Quantum Mechanical Datasets** As real conformations are available in the QM datasets, we evaluated PhysChem on both conformation learning and property prediction tasks. On conformation learning tasks, a Distance MSE metric is reported, which sums the squared errors of all pair-wise distances in each molecule, normalizes it by the number of atoms, and then takes averages across molecules. Note that this metric is equivalent to the Conn-$\infty$ loss with $k = \infty$ in Equation (12). On property prediction tasks, the mean-absolute-errors (MAE) for regression targets are reported. When multiple targets are requested (on QM8 and QM9), we report the Multi-MAE metric in [16] which sums the MAEs of all targets (standardized for QM9). Table 2 and 3 show the results. On conformation learning tasks, PhysNet (*s.a.*) displays significant advantages on learning conformations of small molecules. Specifically, the comparison between PhysNet (*s.a.*) and HamEng indicates that directly *learning forces* in neural physical engines may be superior to *learning energies and their derivatives*. With massive data samples (QM9), the *specialist*, PhysNet (*s.a.*), obtains better results than PhysChem; while on datasets with limited samples (QM7), chemical expertise demonstrates its effectiveness. On property prediction tasks, PhysChem obtains state-of-the-art performances on QM datasets. The comparison between PhysChem, ChemNet (*s.a.*) and ChemNet (*m.t.*) shows that the elaborated cooperation mechanism in PhysChem is necessary, as the straight-forward multi-task strategy leads to severe negative transfer. In addition, the results of ChemNet (*real conf.*) and ChemNet (*rdkit conf.*) show that leveraging real (or generated, geometrically correct) conformations of test molecules indeed helps on some datasets, while the elevations are somehow insignificant.

**Physical Chemical & Biomedical Datasets** On LIPOP, FREESOLV, ESOL and COVID19 with no labeled conformations, we used RDKit-generated conformations to satisfy models that requested conformations of training and / or test molecules. Although these generated conformations are less accurate, distance geometries in local structures are generally correctly displayed. We report the rooted-mean-squared-errors (RMSE) for regression tasks, and the multi-class ROC-AUC (Multi-AUC) metric on COVID19. Table 4 shows the results. PhysChem again displays state-of-the-art performances. Notably, as the numbers of samples in physical chemical datasets (LIPOP, FREESOLV

Table 4: Performances of property prediction on LIPOP, FREESOLV, ESOL, and COVID19.

| | Dataset
Metric | Lipophilicity
RMSE | FreeSolv
RMSE | ESOL
RMSE | SARS-CoV-2
Multi-AUC↑ |
|---|---|---|---|---|---|
| w/o conf. | MoleculeNet [35] | *0.655* | *1.150* | *0.580* | – |
| | Attentive FP [36] | 0.589±0.036 | 0.962±0.197 | 0.612±0.027 | 0.695±0.009 |
| | CMPNN [29] | – | *0.808±0.129* | *0.547±0.011* | 0.692±0.007 |
| | ChemNet (*s.a.*) | 0.577±0.016 | 0.730±0.227 | 0.561±0.035 | 0.757±0.012 |
| | **PhysChem** | **0.568±0.014** | **0.692±0.112** | **0.499±0.025** | **0.792±0.013** |
| w/ conf. | 3DGCN [4] | – | *0.824±0.014* | *0.580±0.069* | – |
| | HamNet [16] | *0.557±0.014* | *0.731±0.024* | ***0.504±0.016*** | – |
| | ChemNet (*rdkit conf.*) | **0.532±0.011** | **0.663±0.029** | 0.504±0.023 | 0.795±0.010 |

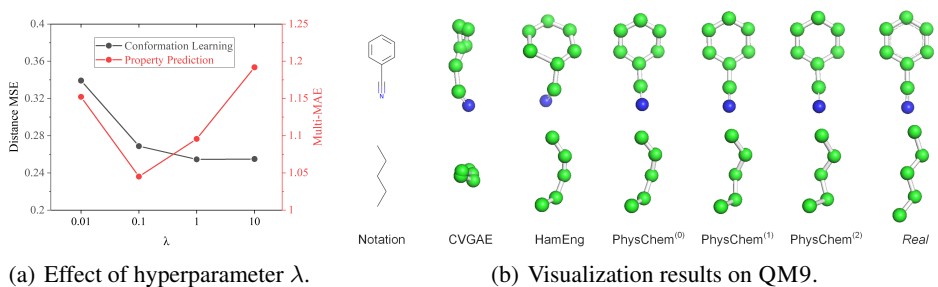

(a) Effect of hyperparameter $\lambda$.      (b) Visualization results on QM9.

Figure 2: Further analyses of PhysChem.

and ESOL) are sparse, the conformation learning task for PhysNet is comparatively tough. This leads to a larger gap of performances between PhysChem and ChemNet (*rdkit conf.*), yet the former is still better than most baselines. The largest elevation of performances is observed on COVID19, which indicates that PhysChem is further competent on more complicated, structure-dependent tasks.

**Further Analyses** Figure 2(a) shows the effect of the loss hyperparameter $\lambda$ on QM9. When $\lambda \leq 0.1$, increasing $\lambda$ benefits both tasks. This indicates that aiding the learning of conformations also helps to predict chemical properties. With larger $\lambda$, the property prediction task is then compromised. Figure 2(b) visualizes the predicted conformations of baselines and PhysNet blocks. Local structures such as bond lengths, angles and planarity of aromatic groups are better preserved by PhysChem.

## 5 Conclusion and Future Work

In this paper, we propose a novel neural architecture, PhysChem, that learns molecular representations via fusing physical and chemical information. Conformation learning and property prediction tasks are jointly trained in this architecture. Beyond straight-forward multi-task strategies, PhysChem adopts an elaborated cooperation mechanism between two specialist networks. State-of-the-art performances were achieved on MoleculeNet and SARS-CoV-2 datasets. Nevertheless, there is still much space for advancement of PhysChem. For future work, a straight-forward improvement is to enlarge the capability of the model by further simplifying as well as deepening the architecture of PhysChem. In addition, proposing strategies to train PhysChem with massive unlabeled molecules is yet another promising direction.

**Broader Impact** For the machine learning community, our work proposes a more interpretable architecture on molecular machine learning tasks and demonstrates its effectiveness. We hope that the *specialist networks* and *domain-related cooperation mechanism* in PhysChem will inspire researchers in a wider area of deep learning to develop novel architectures under the same motivation. For the drug discovery community, PhysChem leads to direct applications on ligand-related tasks including conformation and property prediction, protein-ligand binding affinity prediction, etc. With limited possibility, the abuse of PhysChem in drug discovery may violate some ethics of life science.

**Acknowledgments** This work was supported by the National Natural Science Foundation of China (Grant No. 61876006).

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
