}^{\text{v}} \in \mathbb{R}^{n \times d_{\text{v}}} = (\boldsymbol{x}_1^{\text{v}}, \cdots, \boldsymbol{x}_n^{\text{v}})^\top$ is the matrix of atomic

features, and $X^e \in \mathbb{R}^{m \times d_e} = (x_1^e, \cdots, x_m^e)^\top$ that of bond features. Based on above inputs, PhysNet outputs the dominant conformations of molecules, and ChemNet outputs the representations as well as the predicted chemical / biomedical properties of molecules.

**Overview** Figure 1 is a sketch of PhysChem. An *initializer* first encodes the inputs into initial atom and bond states ($v^{(0)}$ and $e^{(0)}$) for ChemNet, along with the initial atomic positions and momenta ($q^{(0)}$ and $p^{(0)}$) in PhysNet. Subsequently, $L$ PhysNet blocks simulate neural molecular dynamics in the generalized space; $L$ ChemNet blocks conduct geometry-aware message-passing for atoms and bonds. Between each couple of blocks, implicit conformations ($q$s) and states of chemical bonds ($e$s) are shared as expertise. At the top of PhysNet, a *conformation decoder* transforms implicit atomic positions into the 3D Euclidean space; of ChemNet, a sequence of *readout layers* aggregate atom states into molecular representations, based on which molecular properties are predicted with task-specific layers. Notably, two specialist networks are jointly optimized in PhysChem.

### 3.2 Initializer

Given a molecular graph $\mathcal{M} = (\mathcal{V}, \mathcal{E}, n, m, X^v, X^e)$, the initializer generates the initial values for both PhysNet and ChemNet variables. In the initializer, we first encode the input features into initial atom and bond states with fully connected layers, i.e.

$$v_i^{(0)} = \text{FC}\left(x_i^v\right), \ i \in \mathcal{V}; \qquad e_{i,j}^{(0)} = \text{FC}\left(x_{i,j}^e\right), \ (i,j) \in \mathcal{E}. \tag{2}$$

We then adopt the initialization method in [16] to generate initial positions ($q^{(0)}$) and momenta ($p^{(0)}$) for atoms: a bond-strength adjacency matrix $A \in \mathbb{R}^{n \times n}$ is estimated with sigmoid-activated FC layers on bond features, according to which a GCN captures the chemical environments of atoms (as $\tilde{v}$); an LSTM then determines unique positions for atoms, especially for those with identical chemical environments (carbons in benzene, for example). Denoted in formula, the initialization follows

$$A(i,j) = \begin{cases} 0, & (i,j) \notin \mathcal{E} \\ \text{FC}_{\text{sigmoid}}\left(x_{i,j}^e\right), & (i,j) \in \mathcal{E} \end{cases}, \qquad \tilde{V} = \text{GCN}\left(A, V^{(0)}\right), \tag{3}$$

$$\left\{ \left(q_i^{(0)} \oplus p_i^{(0)}\right) \right\} = \text{LSTM}\left(\{\tilde{v}_i\}\right), \quad i \in \mathcal{V} \tag{4}$$

where $V^{(0)} = \left(v_1^{(0)}, \cdots, v_n^{(0)}\right)^\top$ and $\tilde{V} = (\tilde{v}_1, \cdots, \tilde{v}_n)^\top$ are the atom states. The order of atoms in the LSTM is specified by the canonical SMILES of the molecule.

### 3.3 PhysNet

Overall, PhysNet simulates the dynamics of atoms in a generalized, $d_f$-dimensional space ($d_f \geq 3$). In a molecule as a classic dynamical system, atoms move following Newton's Second Law:

$$\mathrm{d}q/\mathrm{d}t = p/m, \qquad \mathrm{d}p/\mathrm{d}t = f, \tag{5}$$

where $q$, $p$ and $m$ are the position, momentum and mass of an atom, and $f$ is the force that the atom experiences. With uniform temporal discretization, the above equations may be approximated with

$$q_{s+1} = q_s + p_s \tau/m, \qquad p_{s+1} = p_s + f_s \tau, \qquad s = 0, 1, 2, \cdots \tag{6}$$

where $s$ is the index of timestamps and $\tau$ is the temporal interval. The calculations in PhysNet simulate such a process. Parameters of PhysNet blocks learn to derive the forces $f$ from intermediate molecular conformations and states of chemical bonds. Correspondingly, two types of forces are modeled in PhysNet, namely the *positional forces* and the *torsion forces*.

**Positional Forces** The positional forces in PhysNet model the gravitations and repulsions between pairs of atoms. In conventional molecular force fields, these forces are generally modeled with (negative) derivatives of *Lennard-Jones potentials*. We therefore propose a similar form for the positional forces taking atomic distances as determinants:

$$f_{j,i}^{\text{pos}} = (r_{j,i}^{-2} - r_{j,i}^{-1}) m_j m_i d_{j,i}, \quad r_{j,i} = \|\text{FC}\left(q_i - q_j\right)\|, \quad i, j \in \mathcal{V}, \quad i \neq j, \tag{7}$$

where $d_{j,i} = \frac{q_i - q_j}{\|q_i - q_j\|}$ is the unitary directional vector from atom $j$ to $i$. Instead of using $l_2$ distances in the generalized space, we use parameterized *interaction distances* $r_{j,i}$ to estimate the forces, which

increase the capability of the network. Here, $r^{-2} - r^{-1}$ approximates the landscape of derivatives of Lennard-Jones potentials[5] with lower-degree polynomials. The approximation is made to avoid numerical issues of using high-degree polynomials, most typically, the gradient explosion.

**Torsion Forces** The torsion forces model the mechanical effects of torsions in local structures. The torsion forces are defined between pairs of atoms that are directly connected by chemical bonds. Estimating local torsions only with positions and momenta of atoms is somehow suboptimal, as the network lacks prior knowledge of chemical characteristics of the bonds (default bond lengths, for example). Alternatively, we leverage the bond states $(e_{i,j})$ in ChemNet as chemical expertise to derive the torsion forces. Specifically, the torsion forces $f^{\text{tor}}$ are calculated as

$$f_{j,i}^{\text{tor}} = w_{j,i} d_{j,i}, \quad w_{j,i} = \text{FC}\left(e_{j,i}\right), \quad (i,j) \in \mathcal{E}. \tag{8}$$

As the torsion forces model chemical interactions, we do not explicitly incorporate atomic mass into the calculation. Notably, atomic information is implicitly considered in the torsion forces, as the bond states integrate atom states at both ends in ChemNet (see the next subsection).

**Dynamics** After estimating the positional and torsion forces, PhysNet simulates the dynamics of atoms following Equation (6). In the $l$-th block of PhysNet, $S$ steps of dynamics are simulated:

$$q_i^{(l,s+1)} = q_i^{(l,s)} + \frac{\tau}{m_i} p_i^{(l,s)}, \quad p_i^{(l,s+1)} = p_i^{(l,s)} + \tau f_i^{(l,s)}, \quad s = 0, 1, \cdots, S-1. \tag{9}$$

Here, $q_i^{(l,0)} = q_i^{(l)}, q_i^{(l+1)} = q_i^{(l,S)}$ ($p$ similarly). $f_i$ is the resultant force on atom $i$, i.e.

$$f_i = \sum_{j \in \mathcal{V}} f_{j,i}^{\text{pos}} + \sum_{(i,j) \in \mathcal{E}} f_{j,i}^{\text{tor}}. \tag{10}$$

**Conformation Decoder and Loss** We use a simple linear transformation to decode implicit atomic positions (and momenta) from the generalized space into the real 3D space:

$$\hat{R} = Q W_{\text{dec}}, \quad W_{\text{dec}} \in \mathbb{R}^{d_f \times 3}, \tag{11}$$

where $Q = (q_1, \cdots, q_n)^\top \in \mathbb{R}^{n \times d_f}$ is the position matrix in the generalized space, and $\hat{R} = (\hat{r}_1, \cdots, \hat{r}_n)^\top \in \mathbb{R}^{n \times 3}$ that of the predicted 3D conformation. We further propose Conn-$k$ ($k$-hop connectivity loss), a spatially invariant loss that supervises PhysNet based on local distance errors: if $C^{(k)} \in \{0,1\}^{n \times n}$ is the $k$-hop connectivity matrix[6] and $D$ is the distance matrix[7], then the $k$-hop connectivity loss is defined as

$$L_{\text{Conn-}k}(\hat{R}, R) = \left\| \frac{1}{n} \hat{C}^{(k)} \odot \left( \hat{D} - D \right) \odot \left( \hat{D} - D \right) \right\|_F, \tag{12}$$

where $\odot$ is the element-wise product, $\|\cdot\|_F$ is the Frobenius norm, $(D, \hat{D})$ are distance matrices of the real and predicted conformations $(R, \hat{R})$, and $\hat{C}^{(k)}$ is the normalized $k$-hop connectivity matrix[8]. The total loss of PhysNet is defined as the weighted sum of Conn-3 losses on all timestamps:

$$L_{\text{phys}} = \frac{1}{Z} \sum_{l,s} \eta^{LS-(lS+s)} L_{\text{Conn-3}}^{(l,s)}, \quad l = 1, 2, \cdots, L, \quad s = 1, 2, \cdots, S, \tag{13}$$

where $\eta < 1$ is a decay factor and $Z = 0.01$ is an empirical normalization factor.

### 3.4 ChemNet

ChemNet conducts message-passing for both atoms and chemical bonds. Besides generating messages with atom states, ChemNet also considers local geometries including bond lengths and angles to adequately characterize local chemical environments. Specifically, in each ChemNet block, *triplet descriptors* are established in the atomic neighborhoods. Atoms merge relevant triplet descriptors to

---

[5]$U^{\text{L-J}} = ar^{-12} - br^{-6}, \quad f^{\text{L-J}} = -\frac{\partial U^{\text{L-J}}}{\partial x} = \left(12ar^{-11} - 6br^{-5}\right)\frac{\partial r}{\partial x}$

[6]$C_{i,j}^{(k)} = 1$ if and only if atoms $i$ and $j$ are $k$- or less-hop connected on the molecular graph.

[7]$D_{i,j} = \|r_i - r_j\|$, where $r \in \mathbb{R}^3$ are 3D coordinates of atoms.

[8]$\hat{C}^{(k)} = L^{-1/2} C^{(k)} L^{-1/2}$, where $L$ is the diagonal degree matrix with $L_{i,i} = \sum_j C_{i,j}^{(k)}$.

generate messages, and then aggregate received messages to update their states. Chemical bonds also update their states by aggregating the states of atoms at both ends.

**Triplet Descriptor** The triplet descriptors $t_{i,j,k}$ are descriptive vectors defined on all atom triplets $(i,j,k)$ with $(i,j) \in \mathcal{E} \wedge (i,k) \in \mathcal{E}$:

$$t_{i,j,k}^{(l)} = \text{FC}\left(v_i^{(l)} \oplus v_j^{(l)} \oplus v_k^{(l)} \oplus l_{i,j}^{(l)} \oplus l_{i,k}^{(l)} \oplus