# OpenReview forum: "Deep Molecular Representation Learning via Fusing Physical and Chemical Information"
_NeurIPS.cc/2021/Conference — NeurIPS 2021 Poster_

### Official Review · Reviewer_HEMY · 2021-07-13

**Rating:** 7
**Confidence:** 3

**Summary:**

The authors are interested in the linked tasks of predicting (i) 3D conformations and (ii) chemical/biomedical properties of molecules. They show that by designing a model (PhysChem) that tackles both of these tasks simultaneously, they can often do better on each individual task than when these tasks are considered separately. This is empirically demonstrated on the QM{7,8,9} datasets, the MoleculeNet  Physical Chemistry Benchmark [32], as well as a SARS-CoV-2 dataset [3], where the authors also compare to a variety of older approaches.




### Further Details on the Architecture
To be more specific about the architecture, PhysChem consists of two networks (which share information at each layer and both take in as input the graph representation of a molecule):
1. PhysNet, is used to predict the conformation of a molecule and it does this by first predicting the initial positions of the atoms (using a graph convolutional network (GCN) and RNN) before then updating these positions using the equations of motion under the action of interatomic forces output by a fully connected NN.
2. ChemNet†, is used to predict chemical/biomedical properties of a molecule, and is a new variant of a message passing neural network [9].

† This is a new network, not to be confused with previous networks also called "ChemNet", e.g.
Preuer, K., Renz, P., Unterthiner, T., Hochreiter, S. and Klambauer, G. (2018) ‘Fréchet ChemNet Distance: A Metric for Generative Models for Molecules in Drug Discovery’, Journal of chemical information and modeling, 58

**Ethical Concerns:**

No.

**Limitations And Societal Impact:**

## Limitations
A. My understanding is that the approach proposes several modifications to the approach used in HamNet [16] but it wasn't clear to me which of these are important to obtain the improved results/why these changes have been made. Examples include:
1. PhysNet predicts the forces directly rather than the energies  (as in HamNet [16]).
2. PhysNet obtains back from ChemNet MPNN information (in the form of "torsion forces") -- in HamNet the information only flows from the physical engine to the MPNN.
3. PhysNet uses a different conformation decoder loss (Eqn 13).
4. ChemNet uses a bespoke message passing scheme, relying on messages that are computed from triplets of graph nodes (as opposed to the ordinary use of pairs -- Section 3.4).

It would be good if empirical evidence/better motivation of these choices was provided.

B. At a higher-level, it would be nice if the general idea of sharing information between conformational and chemical property prediction networks is shown to be useful in a way that is agnostic to the specific architectures used for the separate networks. At the moment this contribution seems somewhat muddled up with the contributions of providing new (and somewhat bespoke) network architectures for these tasks.

C. It actually seems (Table 3) that sharing the information from ChemNet into PhysNet actually degrades performance on the larger datasets QM8/QM9. I would be interested to hear why PhysNet cannot ignore this information if it is not helpful? This also seems to be in conflict with the central claim of the paper (that sharing information between these different tasks is helpful)?


## Societal impact
D. The authors include a Broader Impact subsection in section 5. Here the authors state: "With limited possibility, the abuse of PhysChem in drug discovery may violate some ethics of life science." However, I admit to not understanding the specific potential negative societal impacts that they envisage and would have found an explicit example helpful.

E. In this section the authors also state that their approach provides a more interpretable architecture. However, I would have liked more details on what they meant/some evidence backing up such claims.


**Main Review:**

**SUMMARY**
I think the authors tackle an interesting problem, that of producing better representations of molecules by simultaneously training on different (but linked) tasks, and appear to achieve impressive empirical results. PhysNet alone is shown to outperform previous methods for predicting conformations. However, the proposed networks consist of several moving parts, and the architectural decisions seem fairly arbitrary at times; it would have been nice here to have some intuition or empirical/theoretical justification of the choices made (see limitations box below). Therefore, currently I have gone with a borderline acceptance score but am happy to raise this score if this point gets addressed in the rebuttal.



## Originality
i. The authors produce a novel architecture for predicting both molecular conformations and properties at the same time. Overall, the approach bears some similarities to HamNet [16], for instance the use of a neural physical engine to predict conformations along with the idea of a supplementary MPNN-like architecture to produce a molecular fingerprint. However, there are also some key differences (described in lines 81-91) such as the idea of parameterizing the forces directly or the idea of receiving information back from the MPNN into the physical engine. This ultimately results in better empirical results (Tables 1-3).

## Quality

ii. As far as I can tell the submission is technically sound. The claims are supported empirically and state of the art performance is obtained (compared to some relatively recent methods, e.g. [16] at ICLR 2021). However, this is evaluated in terms of MSE/MAE and it would be nice to also have a comparison in terms of number of parameters/compute costs. There is actually some work in this direction in Appendix H, which shows PhysChem is competitive in this regard, but it would be nice to have this extended to the other models and brought into the main paper.


## Clarity

iii. The approach is quite involved but, considering this, I thought that the authors did a good job at explaining the various steps involved. In particular, Figure 1 provides a helpful overall view of the approach and algorithms 1-3 (in the appendix) provide detailed descriptions of the individual steps involved. The authors provide code which I have glanced through. This is helpful but perhaps could be tidied up a little (e.g. there are a lot of top-level files and I am usure which are important, and also installation instructions should be provided in the README.md)

iv.One aspect relating to clarity that could be improved is better motivation of the choices made. Some of the architectural decisions seem a little ad-hoc and it would be nice to have some intuition for them. For instance, I'm confused as to the motivation behind using Lennard-Jones potentials (line 153) and then the subsequent simplification of these with lower-degree polynomials (line 158). It would be nice to have a more detailed background of this choice for a ML audience in the appendix. (I provide other examples of architectural decisions for which I think further explanation could be provided in the limitations box below.)

v. On a much more minor point, the grammar/choice of words could perhaps be improved in a few places. However, this should be easy to fix. Here are a few examples:
    a. Line 66: "Models that leveraged 3D geometries of molecules generally performed better than those simply used graph notations" => "those _that_ simply used graph notations"?
    b. Line 145-146: "and f is the force that the atom is compelled" => "the force that the atom _experiences_."...?
    c. Line 35: "For example, torsions of bond lengths and angles greatly influence the dynamics the of particle systems" => I think the "the" is unnecessary...?


## Significance
vi. The results improve upon state of the art and therefore I believe may be of interest to other researchers or practitioners. Ablation tests are performed to distinguish the performance improvements due to the PhysNet vs ChemNet but I still am a bit confused as to what components of these individual networks are important (see limitations box below).


## Other Miscellaneous Questions to the Authors

vii. For the molecular properties tasks (e.g. Table 3 and 4) it appears much of the gains come down to the architecture of the MPNN network used, rather than the use of conformational information inside this network. Is this fair? Why do you think this network works better than the previous ones compared to?

viii. Could PhysNet be used as an addition to other networks in the MPNN framework?

ix. I was surprised that only 4 time steps were used in estimating the position/momenta of the atoms in PhysNet. This seems rather low (for instance, in HamNet [16] my understanding is that they use the equivalent of 20 such steps)? Do the positions and momenta converge after so few steps?

x. In the main paper (line 263) you state you set $d_f=128$, yet in the appendix (line 50) this is said to actually be 3? Which value did you actually use? I'm also having trouble reconciling the results in Figure S2 (of the Appendix) with Table 2 of the main paper. For instance, (assuming $d_f=3$) Figure S2 states the results on conformation learning on QM7 are ~0.52; however, in Table 2 they are stated at ~0.49. I would be grateful if you could point out where I am going wrong?

xi. In addition to reporting multi-MAE in the main paper it would be nice to have the individual MAEs for the separate tasks in QM9 in the appendix if possible (for consistency with previous work, e.g. [32, Table S9; 33, Table ST5]).

xii. In figure 2, what do the superscripts for the different PhysChem models represent (e.g. 0,1,2)? (sorry if I missed this).

xiii The LSTM (Eq 4) outputs specific atom coordinates, but I believe the ultimate loss is invariant to rotations/translations of these coordinates? Does this mean the LSTM obtains the same loss even if it arbitrarily translates all initial positions by the same amount? Why not fix the first atom's initial position always to the origin?


**Time Spent Reviewing:**

6

---

> ### Author Response · Authors · 2021-08-10
> **Official Reply to Reviewer HEMY**
>
> We greatly appreciate your careful review and detailed suggestions, and we are glad that our paper could be reviewed by conscientious reviewers like you. We appreciate very much your suggestions w.r.t. organization and languages. We are more than happy to follow your suggestions (ii - v) and further improve our paper.
>
> Here are our answers to your questions (vii - xiii, A - E):
>
> - vii. The improvements of proposing ChemNet (s.a.) as a new MPNN is truly remarkable, yet we think in Table 3 & 4 the improvement of incorporating PhysNet expertises is also consistent and significant (ChemNet (s.a.) vs PhysChem, especially regarding to results in Table 4). Given that the inputs of PhysChem and ChemNet (s.a.) are both 2D structures, i.e. no additional info of test molecules is leveraged, we think the advantage of PhysChem is corroborated. (Even when compared with models using confs of test molecules (3DGCN for example), PhysChem is consistently better.)
>
> 	Indeed, additional demand for PhysChem consists of the efforts in getting real training conformations (actually one-shot RDKit confs are good enough, fast and easy) and some extra time costs (~2-3 times, yet still faster than HamNet); while overall this does not harm the applicability of PhysChem in real scenes (3ms/mol).
>
> 	Why is ChemNet better than other MPNNs? We believe the *triplet descriptor* contributes most. ChemNet (s.a.) itself is in fact an AttentiveFP [33] equipped with triplet descriptor and bond info update. Therefore, the comparison between AttentiveFP and ChemNet (s.a.) is equivalent to an ablation study of these two components.
>
> 	The challenge point of designing ChemNet (instead of using off-the-shelf MPNNs) is **how to use the 3D info**, either predicted or real. This is also the motivation of proposing triplet descriptors: we found that the solutions in 3DGCN and DimeNet are apparently unsatisfactory (sometimes they cannot beat 2D models like [33]), so we propose the triplet descriptors to encode the bonds and angles.
>
> - viii. Yes. PhysNet can be used in combination with existing 3D models (3DGCN, DimeNet et al) since it outputs a predicted conformation. But with 2D MPNNs (MPNN [9] or AttentiveFP [33]), one again should design a strategy to incorporate the 3D info (i.e. with something similar to the triplet descriptors).
>
> - ix. In fact the num of equivalent steps is 8 (2 blocks x 4 steps/block). We found that 8 steps is adequate by observing the visualized results, that molecular conformation generally converges after 8 steps (Fig. 2(b)). Final positions and momenta do converge, compared with first several steps.
>
> - x. We used $d_f=3$, and we are truly sorry for the mistake. Initially PhysChem was designed to operate in generalized space with $d_f \gg 3$ just like HamNet [16]; however at later stage of this work we found $d_f = 3$ leads to satisfactory results. It is actually more natural using $d_f = 3$ as it does not introduce redundant degree of freedom. We forgot to corrected this in the main paper, yet noticed it when we composed the appendix (when we can no longer update the main paper). In fact, the results of using $d_f = 128$ is slightly better (see Fig. S2)
>
> 	The inconsistency of the results are due to different replicas of experiments: the Fig S2 experiments about $d_f$ were conducted separately with different random seeds for data split. Note that we reported 0.49 $\pm$ 0.03 in Table 2 and 0.52 lies in the confidence interval (here $\pm$ 0.03 is the std error, and ~ $\pm$ 0.06 should be the 95% CI).
>
> - xi. Thank you for the suggestion. We are now preparing these results.
>
> - xii. They are conformation outputs from different blocks of PhysNet (a total of 2 blocks were used). It is our mistake to miss out the explanation in the figure description.
>
> - xiii. Yes, you are right about this. The LSTM output is spatial-specific while the ultimate loss is fully spatial invariant. Your proposal is definitely doable. The reasons we didn't do so are: i) as we said above, the initial positions & momenta are originally implicit ones with $d_f \gg 3$. Under this case, there is no straightforward way to assign coordinates for the first atom; ii) an interesting thing is that this may be automatically fixed when weight penalty is used: when weight penalty is used (l2 penalty for parameters, for example), a constant shift ("arbitrary translation") does not generally exist. Empirically, the predicted coordinates of centers of molecules tend to approach [0,0,0]. Taking linear layers as an example: when the weights, especially the bias of a linear layer is penalized, the outputs tend to be 0 if the inputs are appropriately scaled.
>
> - A. Here are some observations for the effect of the modules you proposed. We are sorry that we did not include ablation studies inside PhysNet & ChemNet.
> 	1. Learning forces rather than energies allows PhysChem to perform well even when $d_f = 3 \ll 128$, and it contributes most to the performances of PhysNet.
> 	2. The ChemNet-to-PhysNet expertise indeed did not show significant benefits for conformation prediction, but it allows conf loss to flow gradients to bond representations. See C. for more details.
> 	3. This loss is very similar to the ADJ-k loss in HamNet [16]. We used this loss to avoid calculating $\tilde{A}$ and $\tilde{A}^k$ to save the time cost. ADJ-k tends to emphasize distances between near-hop atoms, while Conn-k equally treats distances with hop < k. We did not discuss these issues due to limitation of space. In our experiments, performances of ADJ-k and Conn-k are almost equivalent.
> 	4. The comparison between AttentiveFP and ChemNet (s.a.) may serve as an ablation of the modification in ChemNet, as ChemNet itself uses a very similar architecture to AttentiveFP.
>
> - B. We agree to your point that currently the modification of PhysChem is a mixture of different (perhaps minor) improvements. We are now planning to try an ablation with PhysNet + 3DGCN to test whether the selection of base modules is important for this idea. On the other hand, by proposing PhysChem we not only want to sell the idea of fusing physics and chemistry, but also want to introduce a better application of molecular fingerprints. When it comes to better applications, these minor but improvements are generally inevitable.
>
> - C. We assume you meant results in Table 2? The reason why this information flow from ChemNet to PhysNet cannot be ignored is that it not only helps PhysNet in deriving torsion information, but also requires better bond representations. Note that expertises themselves are differentiable i.e. we did not stop their gradients. This allows the loss of PhysNet to also propagate to ChemNet representations, as well as ChemNet parameters. We understand that we may better illustrate this with an ablation study. We will work into this and present corresponding results in the camera-ready version if the paper should be accepted.
>
> - D. This claim broadly includes possible yet generally unforseen impacts of PhysChem as well as all drug discovery models. Explicit examples of abuse would be virtual screening for discovering toxins, narcotics, ..., for example.
>
> - E. By better interpretability we referred to that the explicit geometries and the physical-chemical interaction used in PhysChem is more rooted in domain concepts of molecular science. Compared with using general ML models (GNNs, GATs, GGNNs ...), this framework is more intuitive for users with corresponding backgrounds.

---

> > ### Comment · Reviewer_HEMY · 2021-08-25
> > **Responding to authors' rebuttal**
> >
> > I am grateful to the authors for their rebuttal!
> >
> > One of the main concerns I have had in the review period were the worries in Reviewer tYCt's initial review about the validity of the results in Table 2/3. However, from the resulting comments and discussion, this now seems resolved and I am impressed that the authors are moving on to test on larger datasets including GEOM -- I look forward to seeing the results!  Therefore, I have increased my initial review score by 1 (updated in the original review).
> >
> > ## Responses to some of the specific points
> > A. I found the discussion here interesting. In particular, how you view the learning of forces as the most important aspect for the improved performance. It would be great if you could quantify these experimental observations in the paper/appendix.
> > C. Yes you are correct, I meant Table 2 here -- sorry for my mistake. However, I don't think I fully understand your response. You're saying it is important to share this information, as although it degrades the performance on the conformation learning tasks (Table 2), it helps PhysChem do better on the property prediction tasks (Table 3)? Hopefully I will understand better from the ablation study you are adding.
> > vii. Thanks for pointing out how AttentiveFP vs ChemNet can be viewed as an ablation study to evaluate the performance of the triplet descriptor; I had not fully appreciated this before.
> > xiii. Thanks again for the discussion here -- I found this useful.

---

### Official Review · Reviewer_2R2G · 2021-07-16

**Rating:** 8
**Confidence:** 4

**Summary:**

This paper presents a novel method called PhysChem for molecular representation learning by fusing physical and chemical information of molecules. PhysChem consists of two cooperating neural networks, one called PhysNet and another called ChemNet. PhysNet learns 3D conformations by simulating molecular dynamics with parametrized forces, while ChemNet is a geometry-aware message passing network. The experiment results for MoleculeNet benchmark and SARS-Cov-2 datasets showed superior performance over the other state-of-the-art baselines.

GNNs or Geometric NNs are mostly separated to 3D-geometry-aware ones used for quantum-calculation surrogates and 2D-abstraction ones for QSARs predicting properties such as toxicity and solvability. Representation learning that integrates these two aspects is very interesting and promising.


**Limitations And Societal Impact:**

The limitations are partly described in the Conclusion section.

**Main Review:**

This is a very interesting paper that integrates two different aspects of molecules that have been often studied separately. The structure of molecules are quite compositional/combinatorial (e.g. bonds, functional groups, fragments, scaffolds, etc), which motivates us to directly input graphs into neural networks as we seen in GNNs; On the other hand, a molecules exhibits its function in 3D physical space, and act as particle systems that follows the laws of (quantum) mechanics, which is physically constrained rather than being handled as a pure combinatorial object.

As shown in recent studies such as [16](ICLR2021) and [26](ICML2021), neural networks alone can potentially simulate molecular dynamics for conformation prediction. It is thus quite natural to expect that these networks capture physical aspects of molecules in 3D. So, the paper's idea is to learn PhysNet and ChemNet simultaneously: PhysNet uses ChemNet features to have torsion forces in chemical bonds, while ChemNet uses bond-length and bond-angle features from PhysNet 3D conformation. Doing like this, PhysChem is applicable to the situations where the task would be 3D-conformation related, but such conformation information is not directly available. (like a recent KDDCup2021 "PCQM4M-LSC" task to predict DFT-calculated properties from 2D molecular graphs?)

Overall I liked the idea, and but here are some comments.

- For "PhysChem (w/o conf.)" in Table 3 and Table 4, the paper uses UFF-based conformation generation of RDKit for supervised training. Are these conformations unique for each molecule?
The RDKit document says that "Since the 2018.09 release of the RDKit, ETKDG is the default conformer generation method", and it looks to be able to generate multiple conformations by running the distance geometry calculation multiple times from different random start points. These multiplicities don't affect the performance of PhysChem?

- In Table 3, I'm not sure why ChemNet (standalone) can be better than or equal to ChemNet (multitask), PhysChem, and other baselines for QM7 and QM9. QM7 and QM9 is basically geometric-aware tasks, for example, energy is changed if the xyz coordinates are changed. In this sense, if we dropped 3d information as in ChemNet(standalone), it can be hard to predict these quantities. Do you have any rational explanations for this?

- Similarly, what makes PhysNet(s.a) better than HamEng? Without torsion forces from ChemNet, apparently PhysNet would have similar-level features, or even simpler and rough approximation of Lennard-Jones potentials. What brings us this improvement over HamEng?

- For ChemNet, the "triplet descriptors" are unique features, but the remaining would be within the scope of the standard customized MPNNs. So what makes this difference, for example, over MoleculeNet? The ChemNet architecture looks like a variant of gated graph neural networks (GGNNs), and how good if we just grab a standard GGNN here (for Table 3 and 4)?

- Is there any risk of leakage by intensive hyper-parameter tuning for PhysNet, ChemNet, PhysChem? How these results are dependent on careful tuning of hyperparameter tuning?

- The name "PhysNet" might be very confusing because there already exists a widely-cited model called "PhysNet" also applied to QM9 prediction.

PhysNet: A Neural Network for Predicting Energies, Forces, Dipole Moments, and Partial Charges (2019)
https://doi.org/10.1021/acs.jctc.9b00181
For related work, check an ICML2021 paper https://arxiv.org/abs/2102.09844


**Time Spent Reviewing:**

4 hours

---

> ### Author Response · Authors · 2021-08-10
> **Official Reply to Reviewer 2R2G**
>
> We are very glad that you liked our work. With regard to your questions, here are some replies:
>
> > Are RDKit supervisions unique for each molecule?
>
> Yes. We generated one conformation for each train molecule and used it as label. An obvious reason for doing so is that we see conformation generation task as a regression task for atom coordinates, and in regression tasks unique labels are conventionally used. We appreciate very much your advice in generating multiple samples of confs from RDKit. We are not sure about the influence of this setup, but we are more than willing to try it.
>
> > Why ChemNet (s.a.) $\ge$ ChemNet (m.t.) (wrt accuracy)?
>
> This result is actually expected. As we mentioned in the paper, ChemNet (m.t.) used a direct multitask-learning framework by adding a conformation prediction head over the hidden representation in ChemNet. This naive implementation in fact lead to *negative transfer*, that this additional conformation prediction task harms the learning (optimization) of the model, or that it is hard to learn the common knowledge between two tasks simply by sharing hidden representations. By showing this result, we intended to shown the necessity of using the more elaborated cooperation mechanism in PhysChem.
>
> > On QM7, why ChemNet (s.a.) = PhysChem?
>
> This is partially because that QM7 is a smaller dataset regarding to both numbers of molecules and of atoms per molecule. Therefore, the results have relatively large variances. On the other hand, we would like to again emphasize that no confs of test molecules are used in these three baselines. The conf learning tasks in ChemNet (m.t.) and PhysChem only pose explicit requests for the models to capture 3D conf from 2D inputs; they do not bring extra information of test molecules.
>
> To further explain for your concern of xyz determining QM9 energies, and models w/ conf do not perform significantly better than w/o conf, a possible explanation is that both energies and xyzs are derived from confs with lowest energies, which should be determined by the 2D structure, i.e. **the 2D inputs are already self-contained: it already encodes both xyzs and energies by quantum physics**, and essentially the job is to model this physical function (very complicated; generally implemented with Density Functional Theory, DFT) with NN (comparatively simple). This is also the reason that MPNNs (totally without 3D info) can learn QM9 tasks.
>
> > Why PhysNet (s.a.) $\ge$ HamEng?
>
> The major reason is that forces were learned in PhysChem, while in HamEng, energies were learned and their derivatives were used as forces. In our experiments, we found that the optimization (i.e. SGD) of learning forces are much easier than learning energies -- shown not only by the difference in performances, but also the converging speed. In another word, although the expressive ability of the two models could be the same, PhysNet (s.a.) makes the optimization (training) tasks a lot easier.
>
> > How good if ChemNet is implemented with GGNN?
>
> Like we mentioned in the paper, ChemNet is indeed an AttentiveFP [33] (essentially GGNN + graph attention (GAT)) with limited alterations. As results in [33] already showed the significant improvements by adding GATs to GGNNs (the contribution of Attentive FP), we would expect a drop in performance if we used standard GGNNs as ChemNet. Also, the comparison between ChemNet (s.a.) and AttentiveFP shows the effect of our alterations in the model, that these modifications (triplet descriptor, bond message passing et al) indeed work even when no 3D info is incorporated.
>
> > How did we tune the hyperparameters?
>
> Although hyperparameters in Table S2 seem very task-specified, most hyperparameters are in fact directly referred from AttentiveFP [33] and HamNet [16] (as we used similar architectures). We actually did not spend much time in hyperparameter tuning. We only did experiments on different learning rates and loss factor $\lambda$. Empirically during our experiments, the model is robust to hyperparameter selection including the number of layers and dimensionalities.
>
> > Name collision of both PhysNet and ChemNet.
>
> We are very sorry to cause the confusion (we just realized that both PhysNet and ChemNet caused name confusion...) and are thinking about renaming the two submodules into *Phys module* and *Chem module*.

---

> > ### Comment · Reviewer_2R2G · 2021-08-25
> > **Thank you for response!**
> >
> > Thank you for the detailed response. Now, most of my original questions are clear.
> >
> > I found what the authors explained was mostly already written in the original manuscript, and it turned out that my confusions were mostly from my vague understanding of the role/intention of each variant: ChemNet(s.a.), ChemNet(m.t.), and PhysNet(s.a).
> >
> > Here are some discussions for potential improvements.
> >
> > **On QM7, why ChemNet (s.a.) = PhysChem?**
> >
> > One remaining concern was *"On QM7, why ChemNet (s.a.) = PhysChem?"*. Actually, these twos are the best results in Table 3. I understand that QM7 is a small dataset and has large variances, but this is also true for datasets such as LIPOP, FREESOLV, ESOL where we see PhysChem >= ChemNet(s.a.).
> >
> > But I also found that the targets of QM7 are 'atomization energies' (energies for breaking all bonds?) that might be able to be predicted only from 2D molecular graphs (On the other hand, many targets of QM9 seemed geometry-dependent properties.)
> >
> > Anyway it's nice to see this part, and I think it fair to also include this QM7 result. "Limitations, and Societal Impact" would be the community effort (from this year?) to communicate fair understandings and facts about the problem rather than showing only irreproducibly dressed-up champion results.
> >
> > **Conformations as regression targets**
> >
> > I understand that this paper needs unique conformations for regression, and it'll be ok as the first step in this direction. However, in general, a molecule can have rotations about single bonds, and thus conformation should be a space, not a single point, by definition of "conformation". This can be a big limitation to handle molecular tasks. QMX might have only small molecules, but for example, this issue would be inevitable in QSAR tasks (e.g. molecules in PubChem3D have multiple conformations in general).
> >
> > Hawkins, Conformation Generation: The State of the Art, *J. Chem. Inf. Model.* 2017, 57, 8, 1747–1756. https://doi.org/10.1021/acs.jcim.7b00221
> >
> > So it'll be nice to describe these issues as the limitations of the proposed approach in the "Limitations" section. As already asked as 1(d) below, reviewers were specifically instructed to not penalize honesty concerning limitations.
> >
> > https://neurips.cc/Conferences/2021/PaperInformation/PaperChecklist
> >
> > and "In general, authors should be rewarded rather than punished for being up front about the limitations of their work and any potential negative societal impact."
> >
> >
> > **2D inputs are self-contained?**
> >
> > One argument on the statement *"the 2D inputs are already self-contained: it already encodes both xyzs and energies by quantum physics"*
> >
> > It should depend on the node and bond features. The interatomic distances calculated from 3D xyzs are often used as an edge feature, for example, but this is clearly 3D information requiring xyz coordinates in 3D. I guess whether 2D or 3D is not that simple for molecular graphs.

---

> > > ### Author Response · Authors · 2021-09-01
> > > **Reply to Reviewer 2R2G**
> > >
> > > Thank you for your further suggestions. We are sorry if we caused any confusion on the results and will improve the explanation on the experimental setup. About the limitations of testing mostly on small molecules and with single confs instead of conformer ensembles, we will add a section to our paper to describe them. Also, the choice of phrase "2D inputs are self-contained" can indeed be improved. We tried to say that generally dominant confs can be implicitly infered from 2D structures, and the associated properties is thus determined.

---

### Official Review · Reviewer_tYCt · 2021-07-16

**Rating:** 6
**Confidence:** 5

**Summary:**

This paper reports the use of a multi-task/model fusion approach to molecular representation learning: one NN learns to generate conformations through Hamiltonian dynamics with learned forces and another NN does supervised message-passing in an established way


**Limitations And Societal Impact:**

Yes

**Main Review:**

PROs
The architecture fusion is novel and seems effective. By making 3D information a label rather than an input the models are more likely to be transferable, even when there are no existing geometries to train on, as is often the case in large chemical libraries or new chemical spaces

Learning forces directly makes energy not conserved, but it's fine for this application since there is no real "energy" being learned, and it's just a pseudo-energy (and pseudo-forces) to reach the desired geometries. I expect learning straightup pseudo-forces will become a standard in the field

The partitioning of pairwise interactions into the modified LJ and torsions coming from the Chemical Net is new and intereseting but it is not clear whether it's useful or important. A many-body MPNN model like MPNN (Schnet) could learn those terms without crosstalks between the models. I'd be intereseted in learning more about the role of the torsional terms, whether the can be ablated or replaced by learnable manybody MPNN terms in the potential and at what cost in performance / cost.

CONs
The architectures themselves are not very innovative.

The RDKit RMSD for QMX data has me very worried. Most papers report around 0.3 RMSD between RDKit and DFT (and it has be low since the QMX geometries typically come from relaxing initial guesses produced with RDKit) (See arXiv:2105.07246v1 or any of the work from Jian Tang, for example) The values in table 2 and 3 are way too high, and raise concerns about how RMSD is being calculated (atom indexing issues?)

What is a "random" guess for a geometry? Just sprinkling atoms in space will sure give more error than 2 A.

The justification for the low-order polynomial "Lennard Jones" is not great, but at the of the day it's an empirical choice, so it's fine.

The baselines for property prediction are a bit old, so it is not clear how the models stack. I'd encourage the authors to refer to more recent architectures for property prediction, including modern equivariant ones.


Small remarks:
"gravitations" for "attractions"

**Time Spent Reviewing:**

5 h

---

> ### Author Response · Authors · 2021-08-09
> **Official Reply to Reviewer tYCt**
>
> We appreciate your insightful suggestions. We believe that your major concern is around the metric of conformation prediction. Here are some explanations:
>
> 1. In fact, we showed **(Atom-Atom) Distance MSE** in Table 2 instead of **RMSD under optimal alignment**: instead of using coodinates-based metric, we evaluated errors of predicted distances among all pairs of atoms. The definition of Distance MSE can be found at **row 273-276** in the main paper.
>
> 	As we normalized the SSE by **number of atoms** instead of **number of pairs**, the absolute value of this metric is indeed confusing. Accordingly, we show the "random guess" results -- it was implemented as you conjectured, and is used not as a baseline but a reference. By the way, we used the former normalizer because in our experiments, the latter normalizer generally favors molecules with more atoms, and the former one is less biased.
>
> 	We used this distance-based metric because distance-based metrics are not subjected to an optimal alignment, and are empirically more robust when two conformations are less similar. In fact, distance-based metrics (e.g. lDDT, local distance difference test [R1]) are now becoming more popular than RMSDs. Quoted from [R1]:
>
> 	> RMSD has several characteristics that limit its usefulness for structure prediction assessment: the score is dominated by outliers in poorly predicted regions while at the same time it is insensitive to missing parts of the model, and it strongly depends on the superposition of the model with the reference structure.
>
> 2. To further justify the results of our model, we still evaluated RMSD of several baselines during the rebuttal period and below are the results:
> | Baseline                       | RMSD        | Distance MSE |
> | ------------------------------ | ----------- | ------------ |
> | Random Guess                   | 1.972       | 2.799        |
> | RDKit                          | 0.805       | 1.920        |
> | CVGAE                          | 1.364       | 1.052        |
> | HamEng                         | 0.826       | 0.418        |
> | PhysChem                       | 0.732       | 0.255        |
> | PhysNet (s.a.)                 | 0.709       | **0.197**    |
> | PhysNet (s.a.) (*Mixed Loss*)  | **0.483**   | 0.218        |
>
> 	Here, *Mixed Loss* indicates that we used a combination of RMSD loss and Conn-3 loss. So why is 0.805 still far from the reported 0.3 by Jian Tang's work? The reason is that in those papers, the conformtion prediction task was interpreted as a generation task, while in our work, it was simplified as a deterministic "regression" task. A matching score (MAT) is used in [R2], which assert that *multiple conformations* are generated for *multiple targets*:
>
> 	$$MAT(S_g, S_r) = \frac{1}{|S_r|} \sum_{R' \in S_r} \min_{R \in S_g} RMSD(R, R')$$
>
> 	The value of this metric is definitely smaller since when $|S_r|=1$ as is in our case,
>
> 	$$MAT(S_g, R') = \min_{R \in S_g} RMSD(R, R')$$
>
> 	In fact in real cases, the min used in above formula is not so reasonable, as real conf R' is not available, and one cannot pick the argmin conf. when multiple confs are generated.
>
> [R1] V Mariani et al. lDDT: a local superposition-free score for comparing protein structures and models using distance difference tests. Bioinformatics. 2013 Nov 1;29(21): 2722-8. doi: 10.1093/bioinformatics/btt473.
>
> [R2] M Xu et al. An End-to-End Framework for Molecular Conformation Generation via Bilevel Programming. arxiv 2105.07246.

---

> > ### Comment · Reviewer_tYCt · 2021-08-16
> > **Need for a Stronger Benchmark**
> >
> > I thank the authors for the clarification. I do not fully concur about "IDDT now becoming more popular than RMSDs", at least in the field of chemistry but I understand the metric and why it's a reasonable choice, particularly in this "regression setting"
> >
> > While this paper outperforms other methods in the simple QMX datasets (which are not really a conformer generation challenge), I am still concerned that this method (and loss function) may not really capture the challenges of conformer generation for larger molecules and may be biased by the tiny size, low symmetry and low flexibility of the QMX molecules. The GEOM dataset [arXiv:2006.05531] is quickly becoming a reference dataset for larger, more challenging molecules. I'd encourage the authors to test their method in this system. The timeline is probably too tight to finish during the discussions, but I'd love to hear the authors' thoughts.

---

> > > ### Author Response · Authors · 2021-08-19
> > > **Reply regards to tests on larger molecules, conformer ensembles generation tasks (GEOM), and QM9 datasets.**
> > >
> > > - **We surely agree that tests on larger molecules add to the creadibility of a model.** We are now moving on to test larger molecules in new datasets including GEOM. Due to the time limit, we consider it as future work. Nevertheless, here are some other thoughts:
> > >
> > > - **We would like to defend the "regressional setting".** It is a simplification of conformation generation, yet we believe it is a meaningful (if not necessary) one. The major usage of our model as we expect is high-throughput 3D-QSAR (i.e. property prediction) for virtual screening (generally for lead molecules which has appropriate physical, chemical & biomedical properties). The main advantage of the ensembles is that one can use them in tasks such as flexible docking to explore for a best initial conformation to derive better results, but in 3D-QSAR cases these ensembles (as far as we believe) are currently auxilary.
> > >
> > > 	In 3D-QSAR, a deterministic result is generally expected, and in most cases, deterministic conformations suffice. (This is also why we used one-shot RDKit confs as supervisions for MoleculeNet tasks -- although they could be less accurate, they suffice to improve the performances.) It is too time-consuming to generate multiple conformations for 3D-QSAR tasks with generative models (especially with methods that use neural force fields and structural optimization), and one can hardly determine which is the best one to use. Even if ensembles are provided, one would take averages over the ensembles (or perhaps their hidden representations) to derive deterministic property prediction logic. Currently to our best, we do not find models that capture conformer ensembles in reasonable ways, nor references claiming that using conformer ensembles are significantly better than single conformations (say GEOM-QM9 vs QM9 on property prediction).
> > >
> > > 	Another motivation of conformation generation is that these ensembles may be able to reflect the distributional info of conformations. However, in GEOM datasets these ensembles were generated with RDKits (n=50) and relaxed with molecular force fields, where 10 conformers with lowest energies from CREST (semi-empirical, faster, yet less accurate DFT) were selected. In short, the ensembles are essentially local minima on FF PESs. It is questioned whether these local minima of empirical PESs appropriately reflect the real distribution of conformations (slight vibrations over a preferred conformation).
> > >
> > > - **Why are results on QM9 benchmark still meaningful?** Lead molecules in drug discovery generally contain less than 11 atoms. They are generally of "tiny sizes, low symmetry (we assume you mean high symmetry, since they are already tiny?) and low flexibility", but they are useful in designing stable drugs. Also, conformations for QM9 molecules are unique, but they are with lowest DFT energies, instead of derived from empirical methods such as FFs and ranked with empirical DFTs (which were used to generate GEOM molecules). We fully understand that it is a necessary tradeoff to scale up and GEOM developers were doing their best, but one may need more evidence to confidently and broadly use these conformations as ground truths.

---

> > > > ### Comment · Reviewer_tYCt · 2021-08-24
> > > > **Update score, hopeful about extended conformer examples**
> > > >
> > > > I appreciate the authors' thorough reply. Overall I think the method is a clear theoretical innovation and provides performance gains for property prediction and should be accepted. The conformer generation examples are somewhat unsatisfying and ultimately challenged by the lack of thorough benchmarks. This is like introducing a sports car and test drive it to go shopping to the supermarket. The reasons for this oversight are unclear, and the justification for choosing easy benchmarks seems post hoc and not supported by a deep understanding of atomistic simulations.
> > > >
> > > > In more detail:
> > > > I understand the value of the regression setting for geometry and the results are mostly correlated with the more traditional metric anyway, so this is not a fairly moot point. Still, I believe it has fundamental flaws that do not come through in the paper, since internal symmetries _do_ require reindexing of equivalent atoms (need alignment and graph-matching), or the fact that a molecule is _by definition_ not a single geometry. Even if docking models or QSAR are typically applied to a single 3D geometry, there is no one-to-one mapping at all, and assuming so represents the underlying nature of the problem poorly.
> > > >
> > > > "deterministic result is generally expected" without a reference or a survey is hard to take at face value. Many methods out there used in industry are stochastic.
> > > >
> > > > "one would take averages over the ensembles (or perhaps their hidden representations) to derive deterministic property prediction logic" Exactly because molecules do exist as ensembles.
> > > >
> > > > I completely agree with the authors that there's no evidence that 4D-QSAR is any better than 3D-QSAR.
> > > >
> > > > However, I am slightly more concerned with the lack of larger more flexible molecules (the ones that actually are hard to model with traditional conformer generation approaches) in the data. I understand it is out of scope to add more benchmarks, but unfortunately, my assessment of the paper suffers because of this lack of convincing evidence.
> > > >
> > > > The authors also seemed quite confused about the nature of data generation in these training datasets, which may have resulted in their choice. GEOM is applicable because the molecules are larger and harder, even if the authors ignore ensembles (as QM9 does) by picking one arbitrary representative (which they also do in their supervision).
> > > >
> > > > "in GEOM datasets these ensembles were generated with RDKits [...] and relaxed with molecular force fields [...]  the ensembles are essentially local minima on FF PESs" That is exactly what QM9 did, but only keeping 1 member. CREST is not a DFT code, it is a metadynamics code that samples conformational space so that all (thermodynamically accessible) local minima **on the xTB PES** are visited,  rather exhaustively (for electronic structure level of theory)/
> > > >
> > > > "these ensembles were generated with RDKits (n=50) and relaxed with molecular force fields, where 10 conformers with lowest energies from CREST (semi-empirical, faster, yet less accurate DFT) were selected" The authors missed the actual expensive simulation part - the metadynamics that is _seeded_ with rdkit confoemers, explores all the xTB PES, and reports 10 xTB conformers. The simulation did exactly the thing the authors (say they were) concerned by - switches from FF to xTB and samples exhaustively.
> > > >
> > > > "It is questioned whether these [...] these local minima of empirical PESs appropriately reflect the real distribution of conformations" Questioned by who? Clearly not this paper, since the role of metadynamics in sampling is not understood, and this point was only made after being reminded that there are more challenging benchmarks out there. "GEOM developers were doing their best" in _not_ sampling the FF PES but rather exploring a more accurate PES and more exhaustively than distance matrix or ETKDG.
> > > >
> > > > "slight vibrations over a preferred conformation" This seems out of place. Vibrations are not reflected anywhere in the paper and are irrelevant to conformers (by definition a conformer is a local minimum, a static point).
> > > >
> > > > "Lead molecules in drug discovery generally contain less than 11 atoms" It is hard to take this statement at face value without any reference when one looks at approved drugs or Lipinski's rule of five (500 Dalton is 25 heavy atoms or so)
> > > >
> > > > "they are useful in designing stable drugs" I do not follow the point about drug stability
> > > >
> > > > "the ensembles are essentially local minima on FF PESs" This is exactly the case for QM9, with an ensemble of 1 (which may or may not be the lowest energy one since only 1 gets refined with DFT), and exactly *not* the case for GEOM since it samples the xTB (not FF) PES exhaustively (n=50 is to select the seed point and n=10 is after visiting thousands of local minima on the xTB PES)
> > > >
> > > > "conformations for QM9 molecules are unique, but they are with lowest DFT energies" They are not, they are at the lowest force field value, then refined with DFT. So they suffer exactly from the sampling problem the authors seem worried about. Poor sampling or poor ranking with FF may have lead to the wrong local minima.
> > > >
> > > > "empirical DFTs" this terminology suggests the authors do not really understand the nature of electronic structure methods, the approximations in semi empirical tight binding DFT, the accuracy of xTB geometries and energies or atomistic simulations for sampling. They can access  https://xtb-docs.readthedocs.io/en/latest/xtbrelatedrefs.html for a list of references that should allay any concerns about the accuracy of semiempirical methods. Still, even if the geometries came from rkdit (as they do at train time), challenging the approach with larger molecules is the way to go.

---

> > > > > ### Author Response · Authors · 2021-09-01
> > > > > **Reply to tYCt**
> > > > >
> > > > > Thank you for your update of score and the thorough explanation on QM9 vs GEOM dataset. We are more familiar with DL and are indeed no experts in MD and DFT. We are sorry that we got something wrong on the basic setup, as we just started to look into GEOM datasets recently during the rebuttal stage without a profound understanding.
> > > > >
> > > > > Below comments are just for clarification:
> > > > >
> > > > > - by *lead molecules* we refered to general starting candidates in drug discovery. They are not drugs, but smaller molecules used as starting points to further design drugs. This may also help to explain the statement *"they are useful in designing stable drugs"*
> > > > > - *"...even if the geometries came from rdkit (as they do at train time), challenging the approach with larger molecules is the way to go..."* These experiments are on MoleculeNet targets (where rdkit confs are used as train labels), and the samples are drug-like, larger molecules.

---

### Official Review · Reviewer_1R5X · 2021-07-21

**Rating:** 5
**Confidence:** 4

**Summary:**

The authors proposed PhysChem network for molecule representation. PhysChem consists of four modules, the initializer, the PhysNet, ChemNet and readout. Specifically, in PhysNet, the position, momentum and mass of an atom are modeled using rules in mechanics. The authors conduct a series of experiments to verify the proposed algorithm.

**Limitations And Societal Impact:**

Yes.

**Main Review:**

1.	There are many previous work on QM9. You can find the results by searching
(1)	https://paperswithcode.com/sota/formation-energy-on-qm9.
(2)	https://paperswithcode.com/sota/drug-discovery-on-qm9
And QM9 has lots of (sub)tasks. I have two questions towards this point:
(i)	In Table 2, 3, 4, do you use the same way to get the training/validation/test data?
(ii)	A comparison with [ref1] should be discussed.

2.	Some ablation/discussion are required:
a.	What if you use Dihedral angle like [15]?
b.	In PhysNet, do we need both Positional Force and Torsion Forces? What if we use only one of them?

3.	I’m a little confused by your results. In Table 2, why not provide the results of PhysChem? In Table 3, w/ conf, what if we use PhysChem? More surprisingly, on QM7, ChemNet (s.a.) achieves the best results even without conf information.

4.	The RDKit generated conformations are not accurate but useful as illustrated in Table 3. What if we start from RDKit and then finetuning using your proposed method?

[ref1] Molecular Mechanics-Driven Graph Neural Network with Multiplex Graph for Molecular Structures, https://arxiv.org/pdf/2011.07457v1.pdf


**Time Spent Reviewing:**

5

---

> ### Author Response · Authors · 2021-08-09
> **Official Reply to Reviewer 1R5X**
>
> We appreciate your review and valuable questions. Here are some points we would like to clarify:
>
> 1. On QMX datasets we used a random 8:1:1 split of datasets, which is consistent to MoleculeNet and most existing baselines. As there are 12 quantum-mechanics related tasks on QM9, we used a multi-label regression head to train ChemNet, and reported a Multi-MAE metric the same as [16]. For more details of the tasks in our paper, you may refer to [32], where a comprehensive description is available. We appreciate the reference you provided, while honestly speaking, there are too many related approaches on molecular property prediction tasks, and we cannot conduct experiments on all of them.
>
>
> 2. Combining dihedral angle info into PhysNet is very much a promising option, yet we did not include it. One of the reasons is that calculating dihedral angles greatly adds to the temporal cost of the model, since each dihedral angle involves 3 bonds and 4 atoms (roughly speaking, the complexity increases from O(n^2) to O(n^3)). We would sure look into ways in conveniently combining these info into our model, but we would like to leave it as future work.
>
>
> 3. We are sorry if the organization of tables lead to any confusion. However,
> - in fact, the performance of PhysChem is already included in Table 2 (see the last line).
> - PhysChem with real conf. in Table 3 was denoted as ChemNet (real conf.). That is, when real confs are available, PhysChem no longer needs PhysNet to predict confs.
> - indeed ChemNet (s.a) showed remarkable performances on QM7, while we should note that:
>     - most of all, this is a special case on QM7, one among three datasets;
>     - QM7 is a relatively small dataset (a subset of QM9);
>     - the variances on QM7 is large, and the advantage ChemNet (s.a) is not significant.
>
>
> 4. Refining RDKit confs is a good suggestion. The reason we did not do so is that the original design of PhysNet operated in a hidden space ($d_f \gg 3$), and that using explicit 3D coordinates lead to negative transfer in previous work [16]. However, since we moved to using (d = 3) in our final experiments and find the results already satisfactory, refining RDKit confs with PhysNet is definitely doable. We would sure make this one of our future work.

---

> > ### Comment · Reviewer_1R5X · 2021-08-19
> > **Reply to author response**
> >
> > (1) Thanks for your clarification. Please confirm that your data split method, and the data you use, is *EXACTLY* the same as your compared method [18,16,32,33]. Please also make sure your method can be reproduced.
> >
> > (2) Thanks for the message about "dihedral angle". However, usually, a small-molecule has tens of atoms, therefore, I think the additional cost of using "dihedral angle" is acceptable.
> >
> > (3) Can you show more details about "PhysNet predicts the forces directly rather than the energies (as in HamNet [16]).", which is asked by Reviewer HEMY?

---

> > > ### Author Response · Authors · 2021-08-19
> > > **Reply to 1R5X**
> > >
> > > 1. We are sure that the issues you mentioned were properly solved. We attached our codes where random seeds were included for reproducing the results. This should resolve your concerns.
> > >
> > > 2. Calculating these angles adds to ~10x running time of geometry calculation, let alone that we have to adapt triplet descriptors to quaterion descriptors which add to ~2-3x complexity (the avg. degree of atom). As far as we are concerned, it is not acceptable.
> > >
> > > 3. See Eqn. (9) in PhysChem and Eqn. (6) in HamNet for the difference. We are not sure if Reviewer HEMY had the same question.

---

> > > > ### Comment · Area_Chair_urcV · 2021-08-24
> > > > **To Reviewer 1R5X**
> > > >
> > > > Reviewer 1R5X:  Please let me know your further comments on the author's 2-round responses. The authors have provided their codes, which hopefully could remove your concerns about the experimental settings and reproducibility. I would encourage you to comment on the technical innovations of the paper, instead of mainly focusing on the experiments.
> > > >
> > > > Thanks
> > > >
> > > > Area Chair

---

> > > > > ### Comment · Reviewer_1R5X · 2021-08-25
> > > > > **Reply**
> > > > >
> > > > > I do not have further comments technical novelty, and it is also OK for me to accept the paper.
> > > > >
> > > > > [Reply to AC and all] The reason that I focused much on experiments is that: There are quite a few papers about modeling molecules, but the settings are not aligned. We should ensure the split of training, validation and test sets are the same as previous work for fair comparison.  The authors claim that On QMX datasets "we used a random 8:1:1 split of datasets" and "We attached our codes where random seeds were included for reproducing the results". Please ensure you use the same seed (for dataset split) as previous work, so that the followers can make a fair comparison in the future. If not, please report and variance of different dataset split in the future.
> > > > >
> > > > > Although PhysNet is fancy, I am not sure whether it is really useful. (A) In Table 4, seems the generated conformations is not as good as RDKit. (B) In Table 3, ChemNet (s.a.) achieved comparable/almost the same performances of PhysChem, which shows PhysNet does not help much. (C) ChemNet (rdkit conf.) achieved slightly worse performance than PhysChem, but within the standard derivation. (D) Why there is no standard derivation on QM9?

---

> > > > > > ### Comment · Area_Chair_urcV · 2021-08-28
> > > > > > **Explanation on the role of PhysNet**
> > > > > >
> > > > > > Dear Authors,
> > > > > >
> > > > > > Would you please provide more information about the variance w.r.t. different random seeds, and whether the dataset split uses the same random seeds for all the baselines? Furthermore, please provide some discussions on whether PhysNet is indeed critical. These will help increase the confidence of our final assessment on your work.
> > > > > >
> > > > > > Thanks
> > > > > >
> > > > > > Area Chair

---

> > > > > > > ### Author Response · Authors · 2021-09-01
> > > > > > > **Reply to AC and Reviewer 1R5X**
> > > > > > >
> > > > > > > Dear Area Chair and Reviewer 1R5X,
> > > > > > >
> > > > > > > With regard to AC's questions:
> > > > > > >
> > > > > > > - The scope of variances on QM9 is approximately $\pm 0.03$, which indicated that all reported differences of performances were comparatively significant (similar variances were reported in HamNet Table 2). We did not reported the variances as we lack the variances we needed for AttentiveFP and DimeNet. Should the reviewer think these results are important, we will further add them to the table.
> > > > > > > - We cannot assure that the random seeds were the same because most of the baselines did not report their seeds of data split. This is actually a long-lasting issue on QM9 where random splits of data were used. However, as the dataset is relatively large, the variances were generally small.
> > > > > > > - As we explained to Reviewer tYCt, the major reason for proposing PhysNet is not to achieve SOTA conformation generation results. Similar to Hamiltonian Engine in HamNet [16], PhysNet is designed to capture the geometrical info of training molecules and fuse these physical expertise into molecular representation. These results are later used in 3D-QSAR.
> > > > > > >
> > > > > > > With regard to Reviewer 1R5X's questions:
> > > > > > >
> > > > > > > - (A) is not exactly correct. Table 4 compared the performaces of property prediction instead of conformation generation. Also, as PhysNet is learned from RDKit confs, one would actually expect that PhysChem $\le$ ChemNet (RDkit). We want to show that learning these confs with PhysNet yields comparative results to using ground true labels (in this case RDKit labels). As RDKit involves FF MD, learning them with PhysNet is faster.
> > > > > > > - (B) is only true on QM7. The comparisons are significant on QM8/9 and MoleculeNet.
> > > > > > > - (C) is actually the same as (A)
> > > > > > > - (D) the scope of variances on QM9 is approximately $\pm 0.03$, which indicated that all reported differences of performances were comparatively significant (similar variances were reported in HamNet Table 2). We did not reported the variances as we lack the variances we needed for AttentiveFP and DimeNet.

---

### Decision · Program_Chairs · 2021-09-27

**Decision:**

Accept (Poster)

**Comment:**

In this paper, the authors proposed a new neural network model called PhysChem for molecule representation. In PhysChem, a PhysNet and a ChemNet interact with each other to facilitate the learning process. The authors conduct a series of experiments to verify the practical effectiveness proposed algorithm. Overall speaking, the reviewers are positive on this paper. They like the intention of the authors to integrate physical and chemical knowledge in the design of the neural network. But on the other hand, the reviewers also raise some concerns, including the experimental setting, the role of PhysNet, etc. The authors did a good job in addressing most of the concerns, and the consensus made among the reviewers is more leaning towards the positive side. Therefore my recommendation is ACCEPT as a poster.